# Blockchain-Enabled Secure Energy Transactions for Scalable and Decentralized Peer-to-Peer Solar Energy Trading with Dynamic Pricing

Jovika Nithyanantham Balamurugan [1], Devineni Poojitha [1], Ramu Jahna Bindu [1], Archana Pallakonda [2], Rayappa David Amar Raj [1], Rama Muni Reddy Yanamala [3], Christian Napoli [5,6] and Cristian Randieri [4,5,*]

1. Amrita School of Artificial Intelligence, Amrita Vishwa Vidyapeetham, Coimbatore 641112, India; cb.sc.u4aie24223@cb.students.amrita.edu (J.N.B.); cb.sc.u4aie24263@cb.students.amrita.edu (D.P.); cb.sc.u4aie24249@cb.students.amrita.edu (R.J.B.); rd_amarraj@cb.amrita.edu (R.D.A.R.)
2. Department of Computer Science and Engineering, National Institute of Technology, Patna 506004, India; ap23csr1r06@student.nitw.ac.in
3. Department of Electronics and Communication Engineering, Indian Institute of Information Technology Design and Manufacturing (IIITD&M) Kancheepuram, Chennai 600127, India; yanamalamunireddy@iiitdm.ac.in
4. Department of Theoretical and Applied Sciences, eCampus University, Via Isimbardi 10, 22060 Novedrate, Italy
5. Department of Computer, Control, and Management Engineering "Antonio Ruberti", Sapienza University of Rome, 00185 Rome, Italy; cnapoli@diag.uniroma1.it
6. Department of Artificial Intelligence, Czestochowa University of Technology, ul. Dąbrowskiego 69, 42-201 Czestochowa, Poland
* Correspondence: cristian.randieri@uneicampus.it

## Abstract

Decentralized energy trading has been designed as a scalable substitute for traditional electricity markets. While blockchain technology facilitates efficient transparency and automation for peer-to-peer energy trading, the majority of current proposals lack real-time intelligence and adaptability concerning pricing strategies. This paper presents an innovative machine learning-driven solar energy trading platform on the Ethereum blockchain that uniquely integrates Bayesian-optimized XGBoost models with dynamic pricing mechanisms inherently incorporated within smart contracts. The principal innovation resides in the real-time amalgamation of meteorological data via Chainlink oracles with machine learning-enhanced price optimization, thereby establishing an adaptive system that autonomously responds to fluctuations in supply and demand. In contrast to existing static pricing methodologies, our framework introduces a multi-faceted dynamic pricing model that encompasses peak-hour adjustments, prediction confidence weighting, and weather-influenced corrections. The system dynamically establishes energy prices predicated on real-time supply–demand forecasts through the implementation of role-based access control, cryptographic hash functions, and ongoing integration of meteorological and machine learning data. Utilizing real-world meteorological data from La Trobe University's UNISOLAR dataset, the Bayesian-optimized XGBoost model attains a remarkable prediction accuracy of 97.45% while facilitating low-latency price updates at 30 min intervals. The proposed system delivers robust transaction validation, secure offer creation, and scalable dynamic pricing through the seamless amalgamation of off-chain machine learning inference with on-chain smart contract execution, thereby providing a validated platform for trustless, real-time, and intelligent decentralized energy markets that effectively address the disparity between theoretical blockchain energy trading and practical implementation needs.

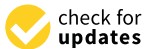

**Keywords:** blockchain; machine learning; Ethereum; smart contracts; energy trading; P2P networks; solar energy prediction; dynamic pricing; Sepolia testnet

## 1. Introduction

The rapid growth of decentralized renewable energy systems [1,2] is essentially transforming the way electricity is produced and consumed. Solar energy, in particular, has enabled consumers to become producers so called *prosumers* who can produce excess energy and directly trade with other parties. However, traditional centralized energy markets face major challenges in efficiently supporting such peer-to-peer transactions due to challenges in predicting generation, implementing dynamic pricing schemes, and ensuring transparent and secure exchanges. The combination of blockchain technology and machine learning offers a strong solution by enabling decentralized, trustless, and intelligent energy trading platforms that respond in real time to changes in supply, demand, and weather conditions.

Blockchain technology facilitates immutable decentralized transactional logging, whereas machine learning offers sophisticated predictive analysis for demand forecasting and real-time price optimization. Furthermore, cross-cutting machine learning applications show how robust, low-latency ML pipelines can already be employed in sensitive, real-time contexts, strengthening the feasibility of the proposed ML+blockchain integration [3]. Despite increasing research interest, numerous existing blockchain-based energy trading platforms are plagued by essential limitations. They encompass reliance on artificial or limited data sets, poor integration among machine learning models and blockchain smart contracts, rigid or static pricing mechanisms, and inadequate integration of real-time weather information for dynamic forecasting. Additionally, scalability bottlenecks, high latency, absence of regulatory clarity, and usability barriers prevent these from reaching their full potential in widespread adoption. Resolving these issues is a must to achieve truly intelligent, scalable, and secure decentralized energy markets.

Blockchain-based energy trading research has progressed through three principal methodologies: security-centric frameworks, smart contract executions, and scalability enhancements, as summarized in Table 1. Research focused on security has primarily aimed at cryptographic safeguarding mechanisms, with certificateless signcryption approaches [4] providing secure P2P energy trading using certificateless encryption and blockchain for privacy and trust, while carbon and energy market implementations [5] utilizing hashed scripts have established blockchain-based decentralized energy and carbon markets using hashed scripts and multisignatures. Nonetheless, these security-centric methodologies are deficient in predictive capabilities and real-time data integration, lacking machine learning integration, dynamic pricing mechanisms, and real deployment scenarios, which constrains their efficacy in dynamic renewable energy markets characterized by fluctuating generation patterns influenced by meteorological conditions and demand shifts.

Smart contract executions have illustrated the viability of automated energy trading via programmable blockchain frameworks. Smart contract-based P2P trading systems [6] developed in Solidity for decentralized energy exchange have demonstrated automated settlement processes, while bilateral trading implementations [7] using Hyperledger Fabric have been tailored for residential energy consumers with two smart trading strategies. Additional investigations include PoET-based energy trading frameworks [8] on Hyperledger Sawtooth with REST APIs and performance analysis, credit-based trading systems [9] on Hyperledger Fabric with penalties for default and secure transactions, and multi-level scoring and non-greedy matching algorithms [10] for fair and efficient blockchain P2P trades.

Game-theoretic methodologies have been examined through Stackelberg game-theoretic markets [11] with smart contracts and dynamic price models, alongside Hyperledger Fabric-based demand response markets [12] using game theory and decentralized pricing, whereas hybrid frameworks [13] have sought to achieve auction and coalition trading mechanisms using Ethereum with real-world data integration. Despite these advancements, all current smart contract implementations suffer from static pricing limitations, a lack of real-time ML forecasting, an absence of external data integration, no smart contract evaluation capabilities, and limited backend integration and automation.

**Table 1.** Literature review of blockchain-based energy trading systems.

| Ref | Approach | Key Features | Limitations |
| --- | --- | --- | --- |
| [4] | Certificateless signcryption on blockchain | Secure P2P energy trading using certificateless encryption and blockchain for privacy and trust. | No ML integration, lacks dynamic pricing, no real deployment |
| [5] | Carbon + energy market using hashed scripts | Blockchain-based decentralized energy and carbon market using hashed scripts and multisignatures. | No predictive modeling, no smart contract evaluation, high market complexity |
| [6] | Smart contract-enabled energy trading | Smart contract-based P2P trading system developed in Solidity for decentralized energy exchange. | Static pricing, no real-time ML forecasting, no external data integration |
| [8] | PoET consensus with Hyperledger Sawtooth | PoET-based energy trading framework on Hyperledger Sawtooth with REST APIs and performance analysis. | No smart contract logic, lacks ML and forecasting integration |
| [7] | Bilateral trading via Hyperledger Fabric | Two smart trading strategies for residential users using Hyperledger Fabric and bilateral contracts. | No ML-driven pricing, lacks forecasting, single-org blockchain |
| [9] | Credit-based energy trading in Fabric | Credit-based trading system on Hyperledger Fabric with penalties for default and secure transactions. | No prediction model, pricing logic lacks real-time automation |
| [10] | Multi-level matching via scoring algorithm | Multi-level scoring and non-greedy matching algorithm for fair and efficient blockchain P2P trades. | No real deployment, lacks weather-based dynamic pricing |
| [12] | Demand-response games using HLF | Hyperledger Fabric-based demand response market using game theory and decentralized pricing. | No ML usage, complex game-theoretic modeling without deployment |
| [11] | Game-theoretic P2P trading with smart contracts | Stackelberg game-theoretic market with smart contracts and dynamic price models in a P2P setup. | No ML forecasting, lacks backend integration and automation |
| [13] | Auction + coalition pricing in Ethereum | Hybrid auction and coalition trading mechanism using Ethereum and real-world Australian data. | No predictive pricing, lacks smart contract to ML feedback loop |
| [14] | Unified blockchain energy market architecture | Unified P2P energy trading architecture using IBFT on Hyperledger Besu integrating 3 energy markets. | High system complexity, no ML prediction, lacks real-time price integration |
| [15] | Polkadot-based scalable carbon-energy trading | Scalable Polkadot-based blockchain system using MILP optimization for energy-carbon markets. | No dynamic ML pricing, lacks on-chain transaction insights |

**Table 1.** *Cont.*

| Ref | Approach | Key Features | Limitations |
|---|---|---|---|
| [16] | Smart contracts for demand response | Ethereum smart contracts for demand response and dynamic pricing using SDR and community validation. | No XGBoost/ML model integration, limited weather automation |
| [17] | Automated DR with game theory and smart contracts | Fuzzy logic-based automated demand response scheduling with smart contracts in local networks. | No real deployment or ML forecasting, static contract control |
| [18] | Security-constrained decentralized trading | ADMM-based decentralized energy trading considering AC power flow and Nash bargaining fairness. | No blockchain layer, lacks real-time automation and ML prediction |
| [19] | Prediction intervals for energy trading | P2P trading using prediction intervals and CVaR risk modeling without blockchain deployment. | No blockchain layer, lacks full-stack automation pipeline |
| [20] | Sharded ABFT blockchain with cross-shard P2P trading | Scalable sharded blockchain with BAC and Hashgraph consensus for fast and secure energy trading. | Focuses on consensus; lacks real-time ML predictions and weather automation |
| [21] | Managing massive renewable energy source integration in hybrid microgrids | Data-driven quad-level approach with adjustable conservativeness for abnormal data handling and renewable integration | Focused on hybrid microgrids; no blockchain or dynamic pricing integration |

Scalability enhancements have been tackled through sophisticated consensus mechanisms and architectural innovations, with sharded Asynchronous Byzantine Fault Tolerance blockchain systems [20] using BAC and Hashgraph consensus for fast and secure energy trading, unified P2P energy trading architectures [14] using IBFT on Hyperledger Besu integrating multiple energy markets, and scalable Polkadot-based blockchain systems [15] using MILP optimization for energy-carbon markets. Research has also explored Ethereum smart contracts [16] for demand response and dynamic pricing using SDR and community validation, alongside fuzzy logic-based automated demand response scheduling [17] with smart contracts in local networks. Security-constrained decentralized trading approaches [18] have employed ADMM-based methodologies for decentralized energy trading considering AC power flow and Nash bargaining fairness, while prediction interval-based trading systems [19] have utilized CVaR risk modeling for P2P trading without blockchain deployment. Additionally, hybrid microgrid integration approaches [21] have addressed massive renewable energy source integration using data-driven quad-level methodologies with adjustable conservativeness for abnormal data handling. However, these scalability-centric solutions invariably compromise real-time data integration and machine learning capabilities, lacking dynamic ML pricing, on-chain transaction insights, XGBoost/ML model integration, weather automation, and real deployment with ML forecasting capabilities. The overarching limitation present in all existing methodologies is the lack of integrated systems that synthesize blockchain security, smart contract automation, machine learning predictions, and real-time environmental data processing into a cohesive platform capable of making intelligent, autonomous energy trading decisions based on weather forecasting and supply–demand optimization.

To address these challenges, this paper proposes an innovative blockchain-based secure energy trading platform that amalgamates Bayesian-optimized XGBoost models with Ethereum smart contracts for dynamic pricing and real-time solar forecasting. Our platform is executed on high-resolution datasets from La Trobe University—[22] the UNISOLAR and UNICON datasets—encompassing over two years of solar generation data at 15 min

intervals from 42 photovoltaic sites, accompanied by synchronized meteorological data including air temperature, dew point, wind status, and solar irradiance.

The key innovation lies in choosing XGBoost over top-of-the-line models like Transformers and other deep learning methods following a stringent comparison. While Transformer models proved successful for renewable energy forecasting, XGBoost proved best suited for blockchain-based energy trading because of four advantages: (1) Computational Efficiency—XGBoost offers sub-50 ms inference times critical for real-time blockchain updates, whereas Transformers take 200–500 ms with complexity in the attention mechanism; (2) Missing Data Robustness—XGBoost employs native surrogate splits to cope with incomplete meteorological data, whereas Transformers severely deteriorate because of missing sensor readings typical in real-world deployments; (3) Data Efficiency—XGBoost offers 97.45% accuracy using 18 months of training data, whereas Transformers typically require much larger datasets (often many year's worth) to perform at similar levels; (4) Interpretability—tree-based feature importance provides better interpretability to achieve compliance with regulatory requirements relative to Transformer's less interpretable attention mechanisms. Our comparative study revealed that XGBoost (97.45% $R^2$) performed better than BiLSTM (95.3% $R^2$) and offered a superior trade-off to Meta-Ensemble approaches, which achieved marginally higher accuracy (99.2% $R^2$) but incurred a 10× higher computational cost, making them impractical for real-time blockchain applications. The Bayesian-optimized XGBoost model is natively implemented with Ethereum smart contracts on the Sepolia testnet and allows dynamic updating of prices autonomously according to weather and demand forecasts. The architecture is made up of ERC-20 tokenization of the units of energy, role-based access control, and robust security features such as reentrancy protection and integrity of cryptographic transactions.

- The work presents a decentralized peer-to-peer (P2P) solar energy trading system on the Ethereum Sepolia testnet. It facilitates direct trading between consumers and prosumers via Solidity-based smart contracts, with guarantees of transparent, automatic, and verifiable execution without the need for a central authority.
- A data-driven XGBoost model optimized with Bayesian optimization is used to predict solar energy production from inputs in the form of real-time meteorological data. The model with an $R^2$ of 97.45% improves market responsiveness by using data-driven, accurate energy forecasts to determine pricing available through a Flask API.
- The system tests and deploys static and dynamic pricing mechanisms. The dynamic pricing scheme tracks the volatility of energy prices based on real-time demand and supply changes, peak demand, and machine learning-driven predictions. Solidity smart contracts control the pricing mechanism with Chainlink oracles providing the verification of external data.
- The platform ensures robust security by the use of SHA-3-based cryptographic hashing, role-based access control (RBAC), and anti-common smart contract vulnerability protection. Reentrancy protection blocks recursive calls, and nonce-based transaction validation inherently blocks replay attacks. Transactions are permanently stored on-chain, and data integrity, traceability, and defense against unauthorized alteration are guaranteed.
- Automated retrieval of weather information is managed by a Python version 3.11. backend with solar forecast forecasts and each 30 min smart contract refresh. Synchronously connected to the Ethereum blockchain with Flask and Web3.py, the system facilitates real-time responsiveness, low-latency communication, and open energy market operation.

## 2. Proposed Decentralized Energy Trading Framework

This section provides a comprehensive methodology for the development of a blockchain-based peer-to-peer grid for peer-to-peer solar energy trading that incorporates machine learning-based energy forecasting, adaptive pricing mechanisms, and cryptographic security standards. The proposed system addresses some of the most significant challenges in peer-to-peer energy markets with a multi-layered approach that leverages advanced predictive analytics, self-executing smart contracts, and real-time data fusion.

### 2.1. System Architecture and Design Framework

The proposed system architecture is based on a five-layer modular model design connected to one another. They include a Data Acquisition Layer exclusively dedicated to capturing real-time weather and energy data, a Machine Learning Layer for predictive and forecast modeling, a Smart Contract Layer for decentralized processing of transactions, a Blockchain Layer for secure record keeping, and an Application Layer responsible for user interfaces and monitoring the system.

As illustrated in Figure 1, the system process begins with the deployment and creation of smart contracts in the Ethereum Sepolia test network using a machine learning model. The machine learning model was initially trained with XGBoost and then improved using Bayesian optimization methods. Subsequently, the backend—implemented with FastAPI—is configured to collect real-time weather data and generate energy price predictions using the trained model. The provided values are then sent to an intellectual delivery agreement on the blockchain, and the complete system operates at a coherent interval of 30 min, ensuring a constant freshness of the data and a specific pricing mechanism.

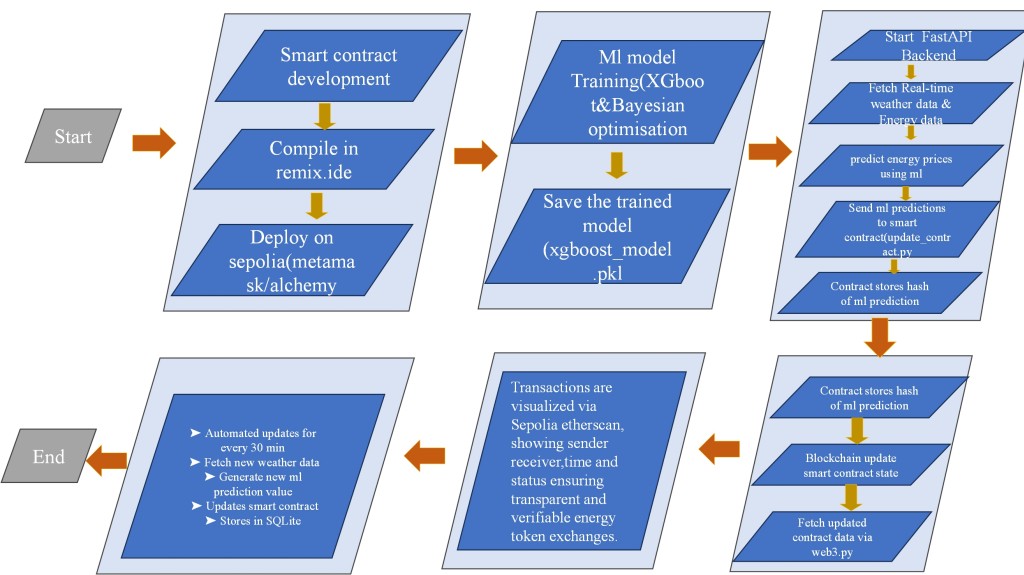

**Figure 1.** Process workflow of proposed energy trading system.

Architectural principles distinguish between fundamental aspects such as scalability, security, compatibility, and resistance to flaws through the architecture of microservices to contribute to free communication between components. This contributes to personal scaling and maintenance of various modules. This contributes to system development and reduces cascade dysfunction across the platform, including effective methods for error management and retirement strategies at each building level. The system is equipped with real-time monitoring and warning possibilities, supporting continuous operation and quick resolution of defects, and with the high accessibility that can be obtained by integrating excess components and load-balancing methods.

### 2.2. Data Collection and Preprocessing Methodology

The system uses a robust data collection methodology using the OpenWeather API, including advanced error management and data verification. Data is collected at 30 min intervals, ensuring temporal consistency and effectively controls API speed limits and actual requirements. Preprocessing includes comprehensive validation procedures, such as anomaly and outlier detection using statistical methods, including Z-score analysis with thresholds of $|z| > 3$, as well as range-based filtering. Missing observations are addressed using time-series–specific imputation methods such as forward filling, backward filling, and seasonal decomposition. Feature Engineering also generates functions of functions such as cyclic coding of variables as a function of time to maintain compatibility along the ladder of measurement using the Min–Max scale. The feature matrix construction follows a systematic approach as defined in Equations (1) and (2), where the input matrix $X \in \mathbb{R}^{n \times d}$ incorporates temperature $T_t$, solar irradiance $I_t$, wind speed $W_t$, humidity $H_t$, cloud cover percentage $C_t$, historical energy generation $P_{t-k:t-1}$ over $k$ previous time steps, temperature gradient $\Delta T_t$, and trigonometric functions $\sin\left(\frac{2\pi h}{24}\right)$ and $\cos\left(\frac{2\pi h}{24}\right)$ to encode the hour of day $h$ for capturing diurnal patterns.

$$X \in \mathbb{R}^{n \times d} \tag{1}$$

$$X = \{T_t, I_t, W_t, H_t, C_t, P_{t-k:t-1}, \Delta T_t, \sin(\frac{2\pi h}{24}), \cos(\frac{2\pi h}{24})\} \tag{2}$$

Historical energy generation data undergo extensive preprocessing with anomaly detection via the Isolation Forest algorithm with a contamination factor of 0.1, seasonal decomposition via STL (Seasonal and Trend decomposition using Loess), data quality evaluation via completeness, consistency, and accuracy metrics, and temporal synchronization with weather statistics via timestamp synchronization protocols. The data preprocessing pipeline also carries out data augmentation processes to balance class and enhance model robustness, and ensure data integrity via checksums and validation against historical patterns,enabling the detection and correction of potential corruption or transmission errors.

### 2.3. Advanced Machine Learning Framework

The module of machine learning uses a holistic approach to compare various algorithms, such as Decision Trees, Random Forest, LightGBM, CatBoost, XGBoost, Bidirectional LSTM, and a model that has a combination of BiLSTM and LightGBM. The performance measures are beyond mere accuracy measures to include computational cost, model explainability, resistance to noisy observations, generalization across changing conditions, and temporal forecasting quality in power systems, where predictability stability under changing environmental conditions is more important than obtaining the highest accuracy scores. Algorithm selection is guided by the particular requirements of energy trading systems, such as the capacity to treat missing meteorological data (common due to sensor failures or communications loss), provide real-time predictions for blockchain use, resist extreme weather conditions, adapt quickly to seasonal changes, comply with regulations, and enable smart grid optimization.

XGBoost was ultimately selected as the base forecasting algorithm due to its superior balance of accuracy, robustness, and efficiency [23]. The algorithm illustrates great ability through its gradient boosting methodology, which repeatedly enhances prediction accuracy by learning from errors in prior models and mapping intricate nonlinear relations between solar energy production and environmental factors. The tree-based framework uses advanced algorithms in dealing with missing values using surrogate splits, uses regularization techniques to prevent overfitting without compromising model flexibility, and offers fine-grained hyperparameter tunability for optimizing across domains. XGBoost

shows a specific advantage in dealing with solar energy data having irregular weather patterns, using its pruning algorithms to remove statistically unimportant branches. The parallel processing architecture greatly improves prediction computation, which is essential for real-time price updates in blockchain energy markets. Bidirectional LSTM, on the other hand, despite having a temporal modeling advantage, is prone to large prediction errors, long training times, and computational complexity that make it inappropriate for real-time applications with limited resources. However, recent studies have shown that compression techniques (e.g., SVD) and acceleration on FPGAs can significantly reduce latency and power consumption for LSTM architectures, making them applicable also in real-time scenarios if properly optimized [24]. The BiLSTM-LightGBM Meta-Ensemble model obtained marginal accuracy gains but added substantial computation overhead and increased prediction latency and integration complexity, which had a specifically negative impact on performance during hours of peak energy prices. Thus, XGBoost emerged as the most effective choice for accurate, efficient, and responsive solar energy forecasting within the proposed framework.

Figure 2 illustrates the comprehensive structure and training process of the XGBoost regressor implemented in the energy prediction system. The left side demonstrates the ensemble architecture of XGBoost, showing how multiple decision trees (Tree-1, Tree-2, Tree-3) are combined through weighted summation to generate the final prediction result. Each individual tree operates on different data subsets and produces residual corrections that address errors from previous trees, enabling the ensemble to capture complex non-linear patterns and interactions in energy generation data. The right side depicts the complete training pipeline, initiated with CSV data inputs containing historical energy generation data, solar irradiation measurements, and meteorological variables, followed by comprehensive preprocessing steps including timestamp standardization, multi-source data integration, and statistical imputation of missing values. The workflow continues with feature engineering to extract temporal patterns and create lagged variables, followed by hyperparameter optimization using Bayesian methods specifically configured for XGBoost parameters. The trained model generates comprehensive evaluation metrics including MAE, MSE, RMSE, and $R^2$, complemented by diagnostic visualizations comparing actual versus predicted values and residual error analysis, ensuring thorough validation of model performance and reliability for energy forecasting applications.

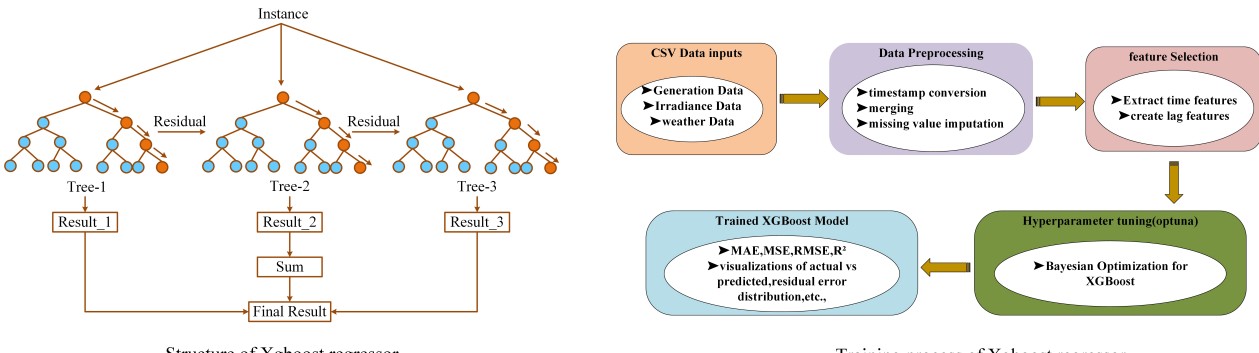

**Figure 2.** Structure and training process of XGBoost regressor.

The optimization objective of the training process is formalized in Equation (3), where $\theta^*$ represents the optimal hyperparameter configuration that minimizes the expected loss function $\mathbb{E}[\mathcal{L}(\theta)]$ over the defined parameter space $\Theta$. The parameter vector $\theta$ encompasses all tunable XGBoost hyperparameters including learning rate ($\eta$), maximum tree depth ($d$), minimum child weight ($\omega$), subsample ratio ($\rho$), and regularization coefficients ($\lambda$, $\alpha$),

while $\Theta$ defines the feasible search boundaries for these parameters based on computational constraints and domain knowledge. The expected loss function $\mathbb{E}[\mathcal{L}(\theta)]$ quantifies the average prediction error across all possible data realizations, ensuring robust model performance under varying environmental conditions and data distributions.

$$\theta^* = \arg\min_{\theta \in \Theta} \mathbb{E}[\mathcal{L}(\theta)] \tag{3}$$

The cross-validation loss function for performance estimation is defined in Equation (4), where $K = 5$ represents the number of temporal cross-validation folds, $N_k$ denotes the sample count in validation fold $k$, $\mathcal{D}_k$ represents the chronologically ordered validation dataset for fold $k$, $Y_i$ is the actual energy generation value (in kWh) for sample $i$, and $\hat{Y}_i(\theta)$ is the model prediction using hyperparameter configuration $\theta$. This formulation computes the mean squared error across all temporal folds, providing an unbiased performance estimate while preventing data leakage through time-series aware validation splits that respect the chronological ordering of energy data, thereby preventing data leakage and ensuring that the model evaluation remains realistic with respect to future forecasting scenarios.

$$\mathcal{L}(\theta) = \frac{1}{K} \sum_{k=1}^{K} \frac{1}{N_k} \sum_{i \in \mathcal{D}_k} (Y_i - \hat{Y}_i(\theta))^2 \tag{4}$$

Bayesian optimization employs the Expected Improvement (EI) acquisition function to intelligently guide hyperparameter search, as formulated in Equation (5). The function $f(\theta^+)$ represents the cross-validation loss at a candidate hyperparameter configuration $\theta^+$, $f(\theta_{\text{best}})$ denotes the best (lowest) loss value observed in previous optimization iterations, and the expectation $\mathbb{E}[\cdot]$ is computed over the Gaussian Process posterior distribution modeling the objective function. The EI acquisition function strategically balances exploration of uncertain hyperparameter regions with exploitation of promising configurations, maximizing the expected improvement over the current best performance while maintaining computational efficiency through informed sampling.

$$\text{EI}(\theta) = \mathbb{E}[\max(0, f(\theta^+) - f(\theta_{\text{best}}))] \tag{5}$$

The hyperparameter search space covers essential XGBoost parameters with domain-specific range to guarantee effective model performance in energy forecasting. The learning rate $\eta \in [0.01, 0.3]$ determines the gradient descent step size, such that smaller values ($\eta \approx 0.01$) ensure conservative learning with more boosting iterations but improved convergence stability, while larger values ($\eta \approx 0.3$) speed up training but may lead to overshooting optimal solutions and oscillations. The size of the deepest tree $d \in \{3, 4, 5, 6, 7, 8, 9, 10\}$ controls the complexity of individual decision trees, shallow trees ($d \leq 4$) minimize overfitting but possibly lead to underfitting in complex energy patterns, and deep trees ($d \geq 8$) model complex feature interactions but exacerbate the risk of overfitting and computational expense. The minimum child weight $\omega \in [1, 10]$ determines the minimum total instance weight needed in leaf nodes, serving as a regularization technique where high values avoid the formation of statistically trivial leaves but can simplify the model too much. The ratio of subsample $\rho \in [0.5, 1.0]$ determines the proportion of training data used for every tree construction, with numbers less than 1.0 adding stochastic regularization to enhance generalization and numbers close to 0.5 potentially causing underfitting from too little training data per tree. The L2 regularization parameter $\lambda \in [0, 10]$ imposes ridge regression penalties to weights of leaves for controlling complexity, and the L1 regularization parameter $\alpha \in [0, 10]$ performs feature selection using lasso penalties. The Bayesian optimization procedure performs 200 evaluation trials for convergence towards near-optimal

hyperparameter settings, each guided by the acquisition function to effectively traverse the high-dimensional space of parameters.

The framework for model validation uses a time-series aware cross-validation approach with `TimeSeriesSplit` and an expanding window approach to avoid temporal data leakage. The validation protocol includes several parts: chronological data splitting with 80% of past data utilized for model training and 20% set aside for temporal testing to preserve the natural sequence of energy generation intervals; 5-fold time-series cross-validation in which each fold $k$ employs all the data before time $t_k$ as the training data and the next temporal section as the validation data, so that realistic forecasting situations are simulated; detailed performance assessment through Mean Absolute Error (MAE) to evaluate average magnitude of prediction error in kWh units, Mean Squared Error (MSE) to penalize large prediction errors more than small ones, Root Mean Squared Error (RMSE) for measuring error magnitude in original energy units, coefficient of determination ($R^2$) for evaluating the proportion of variance in energy generation explained by the model, Mean Absolute Percentage Error (MAPE) to scale-independent error evaluation as percentages, and directional accuracy to evaluate the percentage of correct trend predictions (rising/ falling energy production). Statistical significance testing uses the Diebold–Mariano test, which tests the null hypothesis $H_0$ that the two competing forecasting methods have the same predictive accuracy by calculating the test statistic $DM = \frac{\bar{d}}{\sqrt{\mathrm{Var}(\bar{d})/T}}$, where $\bar{d}$ is the average difference between squared forecast errors, $\mathrm{Var}(\bar{d})$ is the error difference variance, $T$ is the number of forecasting periods, and the test statistic will follow the standard normal distribution under $H_0$.

Model performance is measured in terms of several complementary metrics to present a complete evaluation of prediction accuracy. The Mean Absolute Error (MAE) is given in Equation (6), the Root Mean Square Error (RMSE) in Equation (7), the Mean Absolute Percentage Error (MAPE) in Equation (8), and the coefficient of determination ($R^2$) in Equation (9).

The Mean Absolute Error (MAE) measures the average size of prediction errors in magnitude without directional bias, as expressed in Equation (6). Here, $n$ is the number of test samples in the validation set, $Y_i$ is the true energy production value (in kWh) for the $i$-th observation from solar panel data, and $\hat{Y}_i$ is the associated XGBoost model prediction for the same time interval. The absolute value operator $|\cdot|$ causes overestimation and underestimation errors to be equally weighted in the metric, producing a resilient measure less sensitive to outliers than squared error metrics and having intuitive interpretation in the same physical units as the target variable.

$$\mathrm{MAE} = \frac{1}{n} \sum_{i=1}^{n} |Y_i - \hat{Y}_i| \tag{6}$$

The Root Mean Square Error (RMSE) measures the standard deviation of prediction residuals, as defined in Equation (7). The notation is consistent, with $n$ representing the number of samples, $Y_i$ denoting actual energy values, and $\hat{Y}_i$ representing model predictions. The squared difference $(Y_i - \hat{Y}_i)^2$ amplifies larger prediction errors more heavily than smaller ones, making RMSE more sensitive to outliers and extreme prediction failures compared to MAE. The square root operation $\sqrt{\cdot}$ returns the metric to the original measurement scale (kWh), enabling direct comparison with actual energy generation magnitudes and providing an interpretable measure of prediction uncertainty that is particularly valuable for risk assessment in energy trading applications.

$$\mathrm{RMSE} = \sqrt{\frac{1}{n} \sum_{i=1}^{n} (Y_i - \hat{Y}_i)^2} \tag{7}$$

The Mean Absolute Percentage Error (MAPE) expresses prediction accuracy as a percentage of actual values, as shown in Equation (8). The notation follows previous conventions with $n$, $Y_i$, and $\hat{Y}_i$ representing the sample count, actual values, and model predictions, respectively. The relative error ratio $\frac{Y_i - \hat{Y}_i}{Y_i}$ computes the prediction error as a fraction of the actual value, normalizing for differences in energy generation scales across various solar installations. The absolute value operation $|\cdot|$ ensures symmetric treatment of over-prediction and under-prediction errors. Multiplying by 100% converts the dimensionless ratio to percentage form, providing a scale-independent evaluation that enables fair comparison across solar systems of different capacities, from residential rooftop installations to large utility-scale solar farms.

$$\text{MAPE} = \frac{100\%}{n} \sum_{i=1}^{n} \left| \frac{Y_i - \hat{Y}_i}{Y_i} \right| \tag{8}$$

The coefficient of determination ($R^2$) quantifies the proportion of energy generation variance explained by the XGBoost model, as formulated in Equation (9). The numerator $\sum_{i=1}^{n}(Y_i - \hat{Y}_i)^2$ represents the Residual Sum of Squares (RSS), quantifying the total squared deviations between actual energy measurements and model predictions, essentially measuring the variance left unexplained by the model. The denominator $\sum_{i=1}^{n}(Y_i - \bar{Y})^2$ represents the Total Sum of Squares (TSS), where $\bar{Y} = \frac{1}{n}\sum_{i=1}^{n} Y_i$ is the arithmetic mean of actual energy generation values across all test samples, quantifying the total variance in the target variable that would exist if using only the mean as a predictor. The $R^2$ coefficient ranges from 0 to 1, where values approaching 1 indicate superior model performance with the model explaining most of the variance in energy generation, while values near 0 suggest poor performance equivalent to simply predicting the mean value for all observations. An $R^2 = 0.9745$ (97.45%) indicates that the XGBoost model successfully explains over 97% of the variance in solar energy generation, leaving less than 3% unexplained.

$$R^2 = 1 - \frac{\sum_{i=1}^{n}(Y_i - \hat{Y}_i)^2}{\sum_{i=1}^{n}(Y_i - \bar{Y})^2} \tag{9}$$

*2.4. Smart Contract Design and Implementation*

The smart contract implementation adheres to established security best practices and design patterns to ensure robustness and prevent common vulnerabilities. It incorporates multiple security layers, including role-based permissions using OpenZeppelin's AccessControl, ReentrancyGuard for all state-changing functions, comprehensive parameter validation via custom modifiers, the circuit breaker pattern for system-wide emergency stops, and the proxy pattern to support contract upgradeability.

Figure 3 provides an overall account of the blockchain-based peer-to-peer energy trading process and illustrates the relationship among different components of the system. It indicates how the smart contract developed in Remix has been deployed onto Sepolia via MetaMask and Alchemy and communicates with the machine learning model developed using XGBoost and Optuna optimization on Jupyter notebooks. The API acts as a central coordination layer that utilizes the ML model and the OpenWeather API key to retrieve weather data for real-time energy estimates. Smart contracts store ML predictions as hash for verification and upload blockchain with dynamic pricing and trading data. The system supports SQLite databases for storing ML energy price predictions and model precision, while Web3.py supports ease of connectivity with the blockchain for trading energy and real-time operations.

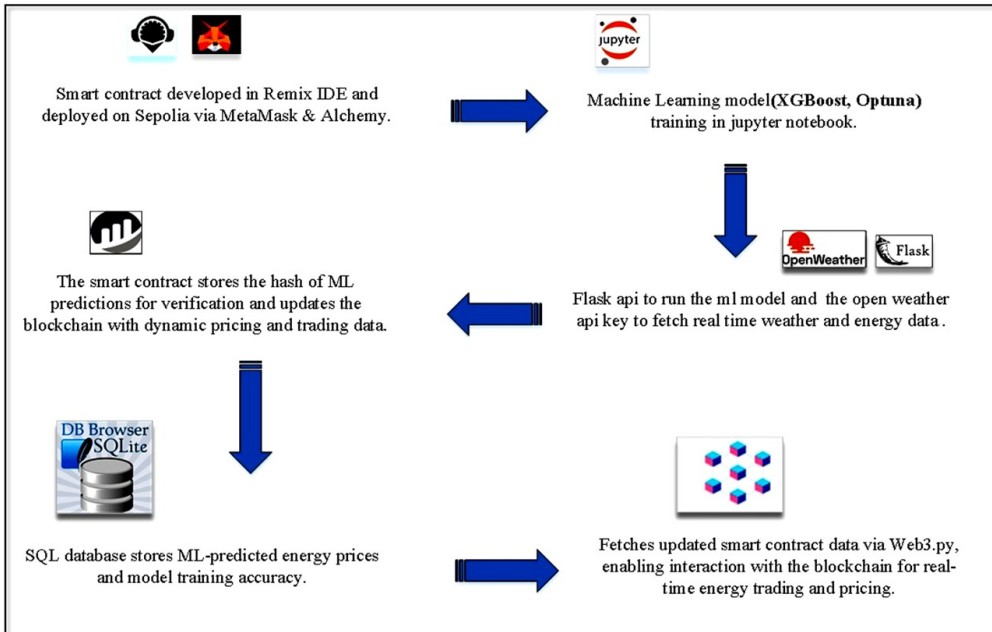

**Figure 3.** Overview of the blockchain based P2P energy trading process.

The dynamic pricing mechanism utilizes a multi-factor model with supply–demand dynamics, temporal fluctuations, and predictive analytics. The calculation of the base price normalizes supply–demand imbalance for avoiding excessive price volatility, as illustrated in Equation (10). The reference base price $P_0$ is the initial market price benchmark denoted in Wei per kWh and forms the building block of the fundamental pricing. The sensitivity parameter $\alpha$ for supply–demand sensitiveness determines the sensitivity of price adjustments to market imbalance, which is generally kept between 0.1 and 0.5 to preserve price stability and yet permit effective market signals. The demand term $D_t$ holds the total energy demand from all the buyers at time $t$, whereas the supply term $S_t$ holds the combined energy offer by all the sellers at the very same moment in time. The regularization factor *epsilon* avoids mathematical singularities when supply tends to zero, maintaining numerical stability in scenarios of extreme market conditions. The ratio $\frac{D_t - S_t}{S_t + \epsilon}$ is a measure of relative supply–demand imbalance, with positive values denoting excess demand and negative values denoting oversupply, having a direct impact on price adjustments via the multiplicative term $(1 + \alpha \cdot \frac{D_t - S_t}{S_t + \epsilon})$.

$$P_t^{\text{base}} = P_0 \left( 1 + \alpha \cdot \frac{D_t - S_t}{S_t + \epsilon} \right) \tag{10}$$

The peak-hour adjustment mechanism accounts for time-of-day effects using an exponential decay function centered around peak consumption periods, as described in Equation (11). The peak hour multiplier coefficient $\beta$ determines the maximum price increase during peak periods, typically ranging from 0.2 to 0.8 to reflect realistic demand-response pricing strategies. The binary peak hour indicator $H_t$ activates the adjustment mechanism during high-demand periods (18:00–22:00), when residential and commercial energy consumption typically peaks. The temporal decay factor $\gamma$ controls how quickly the peak pricing effect diminishes as time moves away from the center of the peak period, with higher values creating sharper price transitions and lower values producing gradual pricing curves. The current time $t$ represents hours elapsed since midnight, while $t_{\text{peak}}$ defines the center of the peak consumption period, typically set at 20:00 h based on grid load profiles. The exponential function $\exp(-\gamma \cdot |t - t_{\text{peak}}|)$ creates a symmetric pricing

curve around the peak time, ensuring that prices gradually increase as the peak period approaches and fall as it recedes.

$$P_t^{\text{peak}} = P_t^{\text{base}} \times \left(1 + \beta \cdot H_t \cdot \exp\left(-\gamma \cdot |t - t_{\text{peak}}|\right)\right) \tag{11}$$

The last dynamic price integrates machine learning prediction confidence as a corrective term, allowing data-driven price optimization according to predicted energy generation, as indicated by Equation (12). The prediction confidence adjustment factor $\delta$ adjusts the impact of ML predictions on final pricing, generally between 0.1 and 0.3 in order to strike a balance between predictive intelligence and market stability. The ML-forecasted solar energy production $\hat{Y}_t$ is the predicted energy output from solar installations at time $t$, computed from the XGBoost model operating on meteorological input like solar irradiance, temperature, and cloud cover. The confidence score $C_t$ is a measure of the model's confidence in its prediction, varying from 0 (very uncertain) to 1 (very confident), calculated from prediction intervals and historical accuracy measures. The maximum supply capacity $S_{textmax}$ sets the theoretical limit of energy production under the best case, as the normalization factor for the correction term. The fractional term $\frac{\hat{Y}_t \cdot C_t}{S_{\text{max}}}$ generates a confidence-weighted prediction ratio that reduces prices in anticipation of abundant energy production when high-confidence predictions anticipate plenty of energy generation, encouraging efficient market clearing and resource allocation.

$$P_t^{\text{final}} = P_t^{\text{peak}} \times \left(1 - \delta \cdot \frac{\hat{Y}_t \cdot C_t}{S_{\text{max}}}\right) \tag{12}$$

The energy allocation process assures equitable energy resource distribution among available resources using utility maximization by balancing with fairness considerations, as expressed in Equation (13). Optimal energy allocation $E_{b,t}^*$ for buyer $b$ during time $t$ is obtained by solving the constrained optimization problem by balancing utility maximization at the individual level with fair system-wide considerations. The number of buyers $B$ is used to denote the number of active users who are interested in purchasing energy during time $t$. The buyer $bs$ utility function $U_b(E_{b,t})$ represents buyer $bs$ satisfaction or economic payoff from the use of energy quantity $E_{b,t}$, often a concave function to represent diminishing marginal utility. The equity weighting factor $\lambda$ modulates the balance between overall system utility and fairness in energy allocation, with larger values favoring equity over efficiency. The Gini coefficient $\text{Gini}(E_{1,t}, \ldots, E_{B,t})$ captures inequality in the allocation of energy among all buyers, from 0 (complete equality) to 1 (absolute inequality), calculated from the cumulative distribution of energy allocations. The optimization goal aims at maximizing the gap between overall utility and penalty of inequality, allowing energy distribution to meet both economic efficiency and social equity principles.

$$E_{b,t}^* = \arg\max_{E_{b,t}} \sum_{b=1}^{B} U_b(E_{b,t}) - \lambda \cdot \text{Gini}(E_{1,t}, \ldots, E_{B,t}) \tag{13}$$

This optimisation problem faces stringent physical and economic constraints that provide realistic and sustainable energy delivery. The supply constraint in Equation (14) guarantees that the aggregate allocated energy among all the buyers cannot surpass the accessible supply $S_t$ and adheres to energy conservation concepts, and inhibits overselling of scarce resources. The demand constraint in Equation (15) provides individual buyer constraints such that no single buyer should be allocated energy beyond the declared request $D_{b,t}$ while ensuring non-negativity of the allocations. The energy allocation variable $E_{b,t}$ symbolises the physical energy amount delivered to buyer $b$ at time $t$ and sits between

zero and the maximum buyer request to offer physically realizable and economically rational distribution patterns.

$$\sum_{b=1}^{B} E_{b,t} \leq S_t \tag{14}$$

$$0 \leq E_{b,t} \leq D_{b,t} \quad \forall b \in \{1, \ldots, B\} \tag{15}$$

*2.5. Blockchain Infrastructure and Cryptographic Security*

The system has been implemented on the Ethereum Sepolia testnet, selected due to its compatibility with the functionalities of the Ethereum mainnet while providing a cost-effective and low-risk environment for the purposes of development and testing. The network is supported by Ethereum's Proof-of-Stake consensus mechanism, which enhances energy efficiency and fosters environmental sustainability. The choice of this particular network was informed by multiple criteria. Among these is its transaction capacity, which stands at approximately 15 transactions per second, deemed adequate for the frequency and volume of energy trading activities. Furthermore, Sepolia's Gas Commission is characterized by predictability and economic feasibility, allowing for accurate economic modeling of transaction expenses. Additionally, the Ethereum Ecosystem offers a well-established suite of tools for community development, documentation, and support, which facilitates optimized integration and deployment. Moreover, the platform exhibits a strong history of security, with resistance mechanisms that mitigate various common types of network attacks.

Figure 4 illustrates the comprehensive workflow of processing blockchain transactions for the energy trading system. The process begins with Step 1: Transaction Creation, whereby key elements of the transactions are gathered, such as Seller ID, Buyer ID, Energy Amount, Final Price, and Timestamp. The transaction attributes are then put through cryptographic hashing in Step 2: SHA-3 Hashing, such that a unique digital fingerprint that serves as a tamper-proof identifier for each energy trading transaction is created. The orientation of numerous transaction hashes using Merkle Root structures, along with the Previous Block Hash to maintain blockchain continuity and security links, and encapsulated with Block Metadata that includes timestamp, nonce, and other crucial block information, is presented in Step 3: Block Creation and Verification. This organized approach ensures the linkage among all the blocks, resulting in an immutable blockchain enabling transparent, verifiable, and secure energy token transactions.

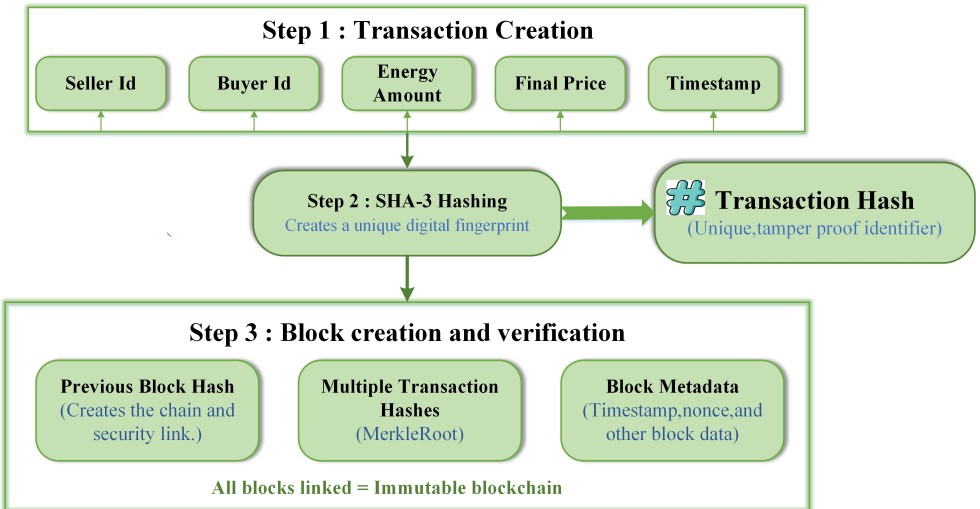

**Figure 4.** Blockchain transaction creation and verification process.

The cryptographic security framework [25] utilizes the SHA-3 hash function to guarantee robust data integrity and authentication of transactions throughout the energy trading platform, as clearly explained in blockchain network security applications. This hash function processes transaction data by handling the entire concatenation of all pertinent transaction fields to yield a distinct cryptographic digest, as outlined in Equation (16). The transaction input $T$ encapsulates the entire transaction structure that includes all critical information necessary for energy trading operations. The variable $n$ signifies the total count of individual fields that make up each transaction, generally comprising seller identification, buyer identification, timestamp, final transaction price, energy quantity, and an anti-replay nonce. Each field element field$_i$ denotes a specific data component within the transaction, with $i$ ranging from 1 to $n$, which includes the seller ID as a unique identifier for the energy provider, the buyer ID serving as the unique identifier for the energy consumer, a timestamp documenting the exact time of transaction creation, the final transaction price $P_t^{\text{final}}$ indicating the agreed-upon cost of energy, the energy amount $E_{b,t}$ representing the quantity of energy being transacted, and the cryptographic nonce that serves to thwart replay attacks. The byte-level concatenation operator $\oplus$ merges all transaction fields in a defined order to form a cohesive input stream for the hash function, thereby ensuring consistent hash generation across various transaction instances. The SHA3 function processes this concatenated input to produce a fixed-length cryptographic hash that acts as a unique digital fingerprint for each energy trading transaction, thus offering tamper evidence and facilitating effective transaction verification while safeguarding sensitive transaction details.

$$H(T) = \text{SHA3}\left(\bigoplus_{i=1}^{n} \text{field}_i\right) \tag{16}$$

The system integrates Merkle tree frameworks for scalable verification of transactions while maintaining cryptographic integrity and reducing computational loads among blockchain users. The Merkle root computation exploits the hierarchical hashing approach to methodically organize many transactions in a single verification anchor as described by Equation (17). The unique transaction tags $T_1, T_2, T_3, T_4, \ldots$ represent the entire set of energy exchange transactions encompassed in a single blockchain block, and each $T_i$ represents a unique buyer-seller energy exchange along with the corresponding transaction details. The hash function $H(\cdot)$ uses the same SHA-3 cryptographic function used for hashing single transactions to hash pairs of transactions and their intermediate results. The byte-wise concatenation operator $\|$ aggregates transaction hashes at each step in the Merkle tree such that the order and content of the transactions play central roles in the computation of the end root hash. The recursive structure begins by pairing the nearest-neighbor transactions $(T_1 \| T_2)$ and $(T_3 \| T_4)$ and computes the respective hash values $H(T_1 \| T_2)$ and $H(T_3 \| T_4)$. The recursive process continues by concatenating these intermediate values and again applying the hash function to obtain the Merkle root. This hierarchical approach reduces the complexity of verification by a factor of $O(n)$ to $O(\log n)$ for inclusion proofs, and the number of transactions aggregated in the block represents the variable $n$. The resulting Merkle root provides a succinct cryptographic summary of all transactions encompassed in the block and allows for the efficient verification of the inclusion of individual transactions without the need for accessing the entire set of transactions and therefore supports lightweight client code and reduced bandwidth requirements for energy trading entities.

$$\text{MerkleRoot} = H(H(T_1 \| T_2) \| H(T_3 \| T_4) \| \ldots) \tag{17}$$

### 2.6. System Integration and API Design

Integration of multiple system elements occurs by RESTful APIs designed using the OpenAPI 3.0 specification. API design follows fundamental stateless software architecture concepts such that request messages contain all context information required and do not rely on the state on the server resulting from previous interactions. The stateless architecture enables system scalability, robustness, and support for horizontal scale-out on multiple server instances. Server computation burden and system responsiveness are optimized by response caching policies through caching the most requested data in the high-capacity memory stores. The API supports layered architecture patterns that enforce rigid separation of concerns among data ingestion, model inferencing, and blockchain interactions, and support maintainable evolution of the system by modules. The uniform interface concept standardizes the identification and manipulation of resources such that understanding APIs and API development processes become easier.

Crucial API routes are introduced to support the basic functions of the platform. They include `POST /api/v1/predict`, which initiates machine learning-based energy generation predictions; `GET /api/v1/prices/current`, retrieving the latest energy prices; `POST /api/v1/transactions`, entering a new energy trading transaction on the smart contract; `GET /api/v1/weather`, providing the latest meteorological information aggregated through various third-party APIs; and `GET /api/v1/analytics`, providing system-level performance statistics as well as usage analytics. Each route incorporates thorough error handling measures, token-based security for robust access control, and rate-limiting functionalities to prohibit abuse while ensuring equal resource allocation among users of the system. The end-to-end architecture for data processing follows a sequential pipeline model that integrates multiple processing phases to transform raw environmental data into executable trading insights.

The pipeline architecture ensures a systematic flow of data through distinguishable computation phases with defined functions and abilities for handling errors, as outlined in Equation (18). The Ingest phase constitutes the data collection processes that gather meteorological information from third-party weather APIs, historical energy production data from solar panel systems, and real-time grid needs data from the utility companies. The Process phase performs extensive data validation to maintain data quality and consistency, runs cleaning processes to remove outliers and handle missing values, and conducts feature engineering to create forecasting variables optimized for machine learning models. The Predict phase deploys the learned XGBoost model to generate predictions for solar energy, computes the confidence intervals for predictions using statistical methods, and checks model predictions using defined operating constraints. The Update phase incorporates the ML predictions into the state modifications of the smart contract, initiates dynamic price changes depending on the predicted supply scenarios, and syncs blockchain data with off-chain databases for auditing. The Notify phase distributes system notifications to registered users through various channels of communication, saves all transactions for meeting regulatory standards, and triggers the alert mechanism when the system identifies anomalies or critical situations.

$$\text{Pipeline} = \text{Ingest} \rightarrow \text{Process} \rightarrow \text{Predict} \rightarrow \text{Update} \rightarrow \text{Notify} \tag{18}$$

### 2.7. Security Framework and Threat Mitigation

The system architecture enforces robust security controls with defense-in-depth mechanisms that secure against attack on multiple architectural layers. On the application level, the system enforces strict input validation and sanitization processes to shield against injection attacks, such as SQL injection protection by way of parameterized queries and cross-site scripting (XSS) prevention to defend user sessions as well as browser contexts.

The authN and authZ system uses industry-standard token-based security features blended with role-based access control mechanisms to provide assurance that only the legitimate user has access to sensitive features.

Security of smart contracts is boosted by formal verification techniques, such as symbolic execution analysis to determine and remove potential vulnerabilities prior to deployment. Denial-of-service attack prevention while remaining cost-efficient in transaction execution is afforded by gas optimization techniques. Integer overflow and underflow attacks are protected against by SafeMath library implementations or more recent Solidity compiler versions that possess built-in overflow protection. The system uses reentrancy guards to avert recursive call attacks that would exploit contract state in an inappropriate manner.

System-level security maintains data confidentiality and integrity using TLS 1.3 encryption protocols for all API communications. Rate limiting and throttling policies manage API access patterns to avoid abuse and ensure equitable resource allocation among valid users. Distributed denial-of-service (DDoS) protection and network segmentation via firewall configurations add other perimeter security layers to the system infrastructure.

Blockchain-level integrity is guaranteed by cryptographic hash chains and sound consensus mechanisms native to the Ethereum network. Private key management adheres to industry standards, such as hardware wallet integration for secure storage of keys and multi-signature rules for high-value transactions to ensure single-point-of-failure mitigation. The system utilizes real-time threat detection features via machine learning algorithms to detect and react to security anomalies in real-time.

The anti-fraud mechanism utilizes a logistic regression model to determine the likelihood of malicious behavior based on behavioral and transactional patterns, as given by Equation (19). The per-user feature values embodied in the vector elements $f_i$ capture distinct behavioral markers ranging from frequency pattern-based transaction indicators detecting abnormally high or low rates of activity, user behavior analytics identifying aberrations in preset interaction patterns, anomaly detection scores based on statistical examination of historical data, geographical position discrepancies which can signal account compromise, and device fingerprinting metrics that monitor hardware and software attributes of accessing devices. The trained weight parameters $w_i$ signify the learned coefficients quantifying the contribution of every feature to fraud likelihood through supervised learning from past fraud transactions and genuine transactions. The model bias term $b$ yields a baseline correction that factors in overall fraud frequency within the energy trading environment, allowing for appropriate calibration of likelihood estimates under varying operational scenarios. The sigmoid activation function $\sigma(\cdot)$ maps the weighted feature linear combination to a bounded probability value between 0 and 1, with values close to 1 representing high fraud probability and those close to 0 reflecting legitimate behavior. This probabilistic score facilitates adaptive threshold-based decision-making for automated fraud detection without prejudice to manual review processes for marginal cases.

$$\text{Fraud\_Score} = \sigma\left(\sum_{i=1}^{n} w_i \cdot f_i + b\right) \tag{19}$$

### 2.8. Performance Optimization and Scalability

The system architecture evolves such that it attains high scalability and performance by virtue of the implementation of a hybrid database model that optimizes techniques for storing and retrieving data in alignment with the particular requirements of a use case. PostgreSQL serves as the master relational database for structured information such as user profiles, transaction records, and system settings which are in need of ACID compliance

and complex relational queries. InfluxDB serves as a time-series database optimized for meteorological readings and energy production values and enables proper storage, downsampling, and retention techniques for high-frequency temporal data. Redis serves as the in-memory datastore enabling very quick caching of user sessions, often evaluated computations, and transitory structures that necessitate microsecond-scale responsiveness. Database optimization techniques consist of index tuning, query-plan-driven optimization, horizontal partitioning of large tables, and connection pools, such that read replicas are employed to distribute the read workloads and enhance throughput. Multiple layers of caching are added through the stack: application-level in-memory caches for objects and computation outcomes, database query result caches aimed at costly SQL queries, CDN edge caching for the static and semi-static API answers, and local caching of smart contract/RPC returns to lower blockchain query latency.

The effectiveness of the caching system is gauged through the cache hit ratio that computes the proportion of requests served from the cache versus those that required exploitation of the underlying data sources, as shown in Equation (20). Here, $Cache_{\text{Hits}}$ is the number of requests served directly from the cache, while $Cache_{\text{Misses}}$ is the number of requests that had to be served by primary data sources (e.g., PostgreSQL, InfluxDB, outside API sources, or smart-contract RPCs). Therefore, the resulting denominator $Cache_{\text{Hits}} + Cache_{\text{Misses}}$ incorporates the overall number of data request accesses during the examined measurement period. The ratio obtained has a decimal value between 0 and 1 such that values approaching 1 represent effective caching and better responsiveness while values near 0 represent ineffective cache exploitation and the need for policy or capacity adjustment.

$$\text{CacheHitRatio} \; = \; \frac{Cache_{\text{Hits}}}{Cache_{\text{Hits}} + Cache_{\text{Misses}}} \tag{20}$$

*2.9. Experimental Design and Evaluation Metrics*

The experimental evaluation of the system follows methodological design, including controlled testing, reliable validation, and large measurement collections. A dataset was used to cover 24 months of historical data on weather and energy, with 18 months being for training and the remaining six months being for temporary verification. To ensure reliability, a high-performance computing environment with accelerated graphics processors is used in five cross-sectional controls to support effective training and evaluation of machine learning models.

Prediction accuracy is assessed using multiple error metrics, including root mean square error (RMSE), mean absolute error (MAE), and mean absolute percentage error (MAPE), defined respectively as shown in Equations (6)–(8).

In addition to model accuracy, system performance is measured using throughput (transactions per unit time), latency defined as the 95th percentile of response time, and availability calculated as shown in Equation (21):

$$\text{Availability} = \left( \frac{\text{Uptime}}{\text{Total\_Time}} \right) \times 100\% \tag{21}$$

Uptime refers to the total amount of time the system spends running and available to users, while the Total_Time represents the entire period to be examined. This measure provides a reflection on the stability and dependability of the system in running.

The economic efficiency is evaluated through cost savings and market efficiency measures, as presented in Equations (22) and (23). The cost savings measure values the economic gains of utilizing dynamic pricing policies instead of conventional static pricing policies in the context of peer-to-peer energy trading systems. The parameter

$P_{\text{static}}$ represents the total monetary cost of the classical static pricing policies, whereby the energy price acts as a constant despite supply and demand alterations, meteorological considerations, or temporal variations during the trading period. The parameter $P_{\text{dynamic}}$ represents the total cost through the innovative intelligent dynamic pricing model that adjusts the prices in real-time using the predictions by the XGBoost machine learning model, the OpenWeather API data, and the prevailing market data recorded on the blockchain. The term $(P_{\text{static}} - P_{\text{dynamic}})$ calculates the absolute difference in cost across the two pricing policies, and the division by the parameter $P_{\text{static}}$ normalizes this difference to a ratio to the base case static cost. The positive values represent the economic gains by the adaptive price system and demonstrate the improved cost optimization through the intelligent price adjustments.

$$\text{Cost\_Savings} = \left( \frac{P_{\text{static}} - P_{\text{dynamic}}}{P_{\text{static}}} \right) \times 100\% \tag{22}$$

The measure of market efficiency considers the degree to which the dynamic pricing mechanism reduces instability in the energy market relative to general market trends, using variance reduction analysis. The parameter Price_Variance measures the statistical variance in energy prices generated by the suggested blockchain-enabled dynamic pricing mechanism, reflecting the patterns of price movements in the regulated peer-to-peer trading system during the evaluation period. By contrast, the parameter Market_Variance represents the variance observed in the larger regional or national energy market prices during the evaluation period and serves as an outside standard for comparison while accounting for intrinsic market instability beyond system control. The ratio $\frac{\text{Price\_Variance}}{\text{Market\_Variance}}$ measures the degree of relative price stability realized by the system such that lower ratio values indicate greater market stability. The subtraction of unity transforms this ratio to represent an efficiency score such that values close to 1.0 represent optimal market effectiveness through maximum price instability reductions relative to outside market trends and demonstrate the system's ability to create more predictable and stable trading environments for the parties operating in decentralized energy markets. In this case, Price_Variance represents the price variance generated by the dynamic pricing system while Market_Variance represents the price variance inherent in the larger energy market prices. Measures approaching 1 represent a higher degree of market effectiveness, such that the system optimally reduces price instability relative to overall market trends.

To be statistically valid, the assessment incorporates two-tailed *t*-tests at a $\alpha = 0.05$ level of significance, computation of Cohen's *d* to gauge effect size measures, and 95% confidence intervals for all point estimates. Bonferroni adjustment is used to control Type I errors in multiple hypothesis testing and ensuring rigorous inference.

$$\text{Market\_Efficiency} = 1 - \left( \frac{\text{Price\_Variance}}{\text{Market\_Variance}} \right) \tag{23}$$

static pricing scheme and $P_{\text{dynamic}}$ is the total cost under the proposed dynamic pricing system. This metric quantifies the percentage reduction in costs achieved through intelligent price optimization.

### 2.10. Ethical Considerations and Regulatory Compliance

The system is built while placing a significant focus on ethical considerations and legal compliance. The security of user data privacy is ensured through the implementation of data processing frameworks in alignment with GDPR such as pseudonymization, anonymization, and the safe storage of data. A consent management system enables users to control their data collection intentions, while explicit data-use policies foster clarity regarding the storage, sharing, and processing intentions of data.

Compliance is achieved through the observance of regional energy market regulations, meticulous maintenance of transaction records to support auditing processes, and adherence to standards pertaining to financial services in relation to payment procedures. The system incorporates consumer protection elements, including the enforcement of equitable pricing, mechanisms for resolving disputes, and transparency in fee disclosure, thereby guaranteeing the fair treatment of consumers and safeguarding their rights.

Promotion of environmental sustainability is achieved through the use of renewable energy sources, carbon footprint tracking, and promotion of initiatives related to the certification of green energy. Environmental compliance audit reporting tools are provided to support the conduct of environmental compliance audits. The platform adheres to social responsibility values through the enhancement of energy access equity, especially through the involvement of marginalized peoples. Multiple awareness and education materials are provided to enlighten users on the merits related to renewable energy. Collectively, the platform helps in the achievement of the United Nations Sustainable Development Goals through the use of blockchain and artificial intelligence technologies to equalize access to clean and reliable energy.

## 3. Results and Discussion

This section deals with smart contract deployment, ML-based solar power forecasting, API integration for energy and weather data, Blockchain contract updates, SQL database logging for predictions and weather data, Transaction logs and verification, static pricing analysis and dynamic pricing analysis, transaction event logs, and security mechanisms.

### 3.1. Smart Contract Deployment and Transaction Analysis

The implementation of the smart contract on the Ethereum Sepolia Testnet establishes the basis for the peer-to-peer solar energy trading system. The smart contract incorporates several essential functionalities, such as active seller monitoring for energy providers, dynamic pricing adjustments based on real-time demand and meteorological conditions, machine learning-driven forecasting for solar energy generation predictions, and real-time weather data acquisition from the OpenWeather API.

The contract was successfully deployed on the Sepolia testnet at the following address:

Currently, the system has one active seller (Listing 1). This active seller monitors fluctuations in energy prices and updates the smart contract upon receiving new machine learning predictions or weather updates.The SecureEnergyTradingPlatform smart contract was successfully deployed on the Sepolia testnet, and its address is provided in (Listing 2).

Currently, the system has one active seller, deployed at the following address:

**Listing 1.** Address of the active seller in the system.

```
0x8cbc1ADD5081900a69BAE4200DF1a716f514d7f0
```

**Listing 2.** Address of the deployed SecureEnergyTradingPlatform contract on Sepolia.

```
0x349C7DDAc8091d6663bac0E14c90cf8E3C594463
```

Table 2 presents the transaction metrics for a typical contract update operation on the Sepolia testnet. The transaction involved no transfer of ETH, as it was intended solely to update on-chain contract states such as energy pricing, seller registration status, and ML-driven predictions rather than facilitate fund transfers. Prices were represented in Wei, the smallest denomination of Ether (1 ETH = $10^{18}$ Wei), to enable high-precision computations. This approach minimizes rounding errors in microtransactions, thereby enhancing the accuracy and stability of the dynamic pricing mechanism implemented within the smart contract.

**Table 2.** Ethereum transaction metrics on Sepolia Testnet.

| Parameter | Value |
|---|---|
| TX Hash | 0xc2c3f2b19b3d6…02a3a3 |
| Block Number | 7,912,476 |
| Confirmations | 16,522 |
| Gas Used | 33,984 (16.99%) |
| TX Fee | 0.00017021 ETH |
| Gas Price | 5.00874 Gwei |
| Wei Used | 217920817994 Wei |
| Contract Address | 0x349C7DDA…E3C594463 |
| Method Called | 0x74c115dd |

*3.2. Machine Learning-Based Solar Energy Prediction*

According to the comparative analysis in Table 3, XGBoost with Bayesian optimization was the optimal choice for predicting energy prices among the eight evaluated machine learning models. The Bayesian-optimized XGBoost model exhibited superior robustness (rated "Very High") and achieved 97% accuracy, demonstrating exceptional performance for energy trading applications. Decision trees, while highly interpretable and computationally efficient, were prone to overfitting and achieved only 94% accuracy. Random Forest and LightGBM showed good generalization capabilities with 96% accuracy but fell short of the gradient boosting methods' precision. CatBoost performed admirably with 97% accuracy and excellent categorical variable handling. The Transformer model, despite its advanced attention mechanisms, achieved only 81.0% accuracy and demonstrated slower training times with moderate interpretability, making it less suitable for real-time energy prediction tasks. Bidirectional LSTM, despite using a deep learning approach, achieved only 95.3% accuracy and underperformed due to limited temporal dependencies in solar data. The Meta-Ensemble model (LSTM + LightGBM) reached the highest accuracy at 99.2% but incurred prohibitively high computational overhead due to its combined sequential and tabular architecture, making it impractical for real-time blockchain applications.

XGBoost emerged as the most suitable choice for blockchain-based energy trading due to four critical advantages that align with the specific requirements of decentralized energy markets. First, computational efficiency enables XGBoost to achieve sub-50 ms inference times, which are essential for real-time blockchain price updates, whereas Transformers require 200–500 ms due to attention mechanism complexity, and deep learning architectures demand significantly more computational resources. Second, missing data robustness allows XGBoost to handle incomplete meteorological data through native surrogate splits, a common challenge in real-world deployments where sensor failures or communication interruptions frequently occur, while Transformers and deep learning models experience severe performance degradation with missing sensor readings. Third, data efficiency enables XGBoost to achieve 97% accuracy using 18 months of training data, whereas Transformers typically require multi-year datasets to reach comparable performance levels, making XGBoost more practical for emerging renewable energy installations with limited historical data. Fourth, interpretability through tree-based feature importance provides transparent decision-making processes essential for regulatory compliance and market trust, whereas Transformer attention mechanisms and deep learning models offer limited explainability for energy trading decisions. XGBoost's hyperparameter optimization through Bayesian techniques ultimately yielded the most reliable and consistent predictions under varying environmental conditions, making it the ideal choice for the dynamic and resource-constrained nature of blockchain-based energy price forecasting.

**Table 3.** Detailed parameters of ML models for energy price prediction.

| Parameter | Value |
| --- | --- |
| Model | Decision Tree |
| Accuracy (%) | 94 |
| Training Time | Fast |
| Interpretability | High |
| Robustness | Moderate |
| Remarks | Simple but may overfit; useful for rule-based insights |
| Model | Random Forest |
| Accuracy (%) | 96 |
| Training Time | Moderate |
| Interpretability | Moderate |
| Robustness | High |
| Remarks | Ensemble of trees; reduces overfitting; good generalization |
| Model | LightGBM (LGBM) |
| Accuracy (%) | 96 |
| Training Time | Very Fast |
| Interpretability | Moderate |
| Robustness | High |
| Remarks | Efficient with large data; handles categorical features well |
| Model | CatBoost |
| Accuracy (%) | 97 |
| Training Time | Moderate |
| Interpretability | Moderate |
| Robustness | High |
| Remarks | Excellent accuracy; native support for categorical variables |
| Model | Bidirectional LSTM |
| Accuracy (%) | 95.3 |
| Training Time | Slow |
| Interpretability | Low |
| Robustness | Moderate |
| Remarks | Deep learning approach; underperforms due to limited temporal dependencies |
| Model | Meta-Ensemble (LSTM + LightGBM) |
| Accuracy (%) | 99.2 |
| Training Time | Very Slow |
| Interpretability | Low |
| Robustness | Very High |
| Remarks | Combines sequential and tabular learning; high computational overhead |
| Model | Transformer |
| Accuracy (%) | 81.0 |
| Training Time | Moderate |
| Interpretability | Medium |
| Robustness | High |
| Remarks | Slower training and less interpretable |
| Model | XGBoost (Bayesian Optimized) |
| Accuracy (%) | 97 |
| Training Time | Slow |
| Interpretability | Moderate |
| Robustness | Very High |
| Remarks | Highly accurate; best for performance-critical tasks; tunable |

Using an XGBoost Regressor model, the system forecasts solar energy generation based on historical trends and current weather data. The model is trained with key parameters like wind speed, air temperature, cloud opacity, and solar irradiance. As depicted in Figure 5, the actual versus predicted solar energy generation demonstrates the model's high predictive accuracy, with predicted values closely aligning with the actual measurements. Figure 6 illustrates the feature importance analysis of the XGBoost model,

revealing the relative contributions of different input features to the energy generation predictions. The residual error distribution is presented in Figure 7, providing insights into the model's prediction errors and their statistical characteristics. Figure 8 shows the temporal progression of solar energy generation, highlighting the model's ability to capture time-dependent variations in energy output.

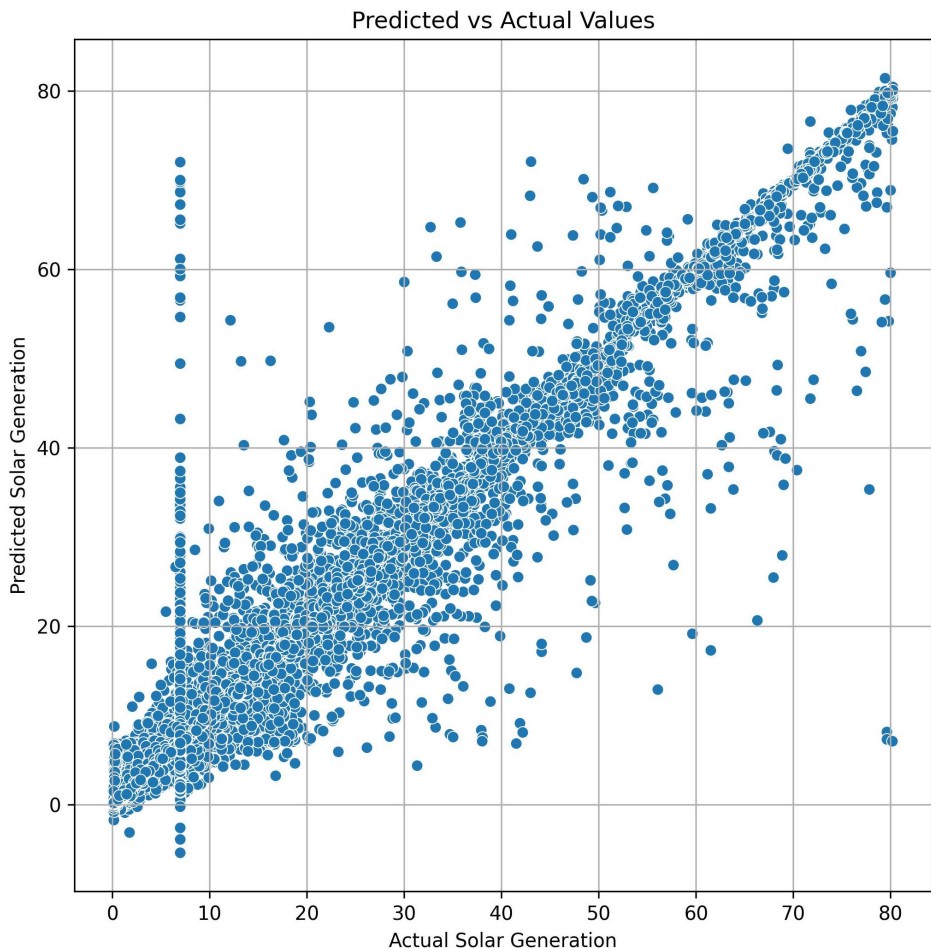

**Figure 5.** Actual vs. predicted solar energy generation.

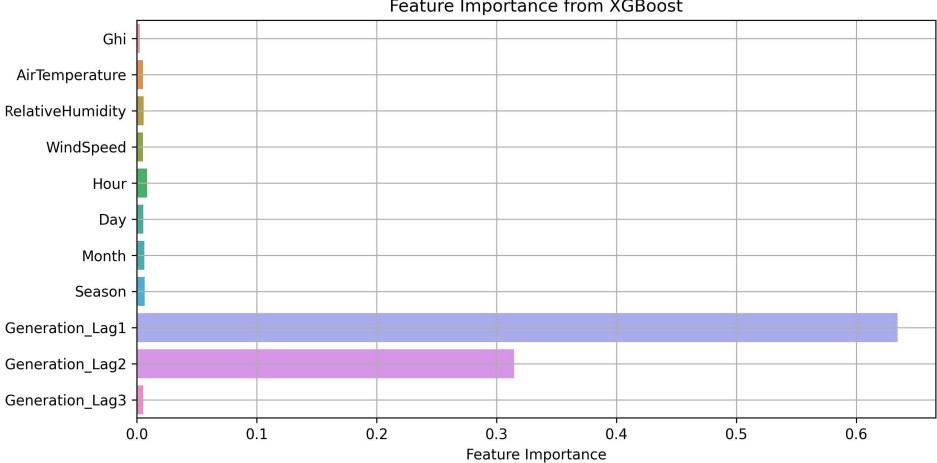

**Figure 6.** Feature importance analysis using XGBoost model.

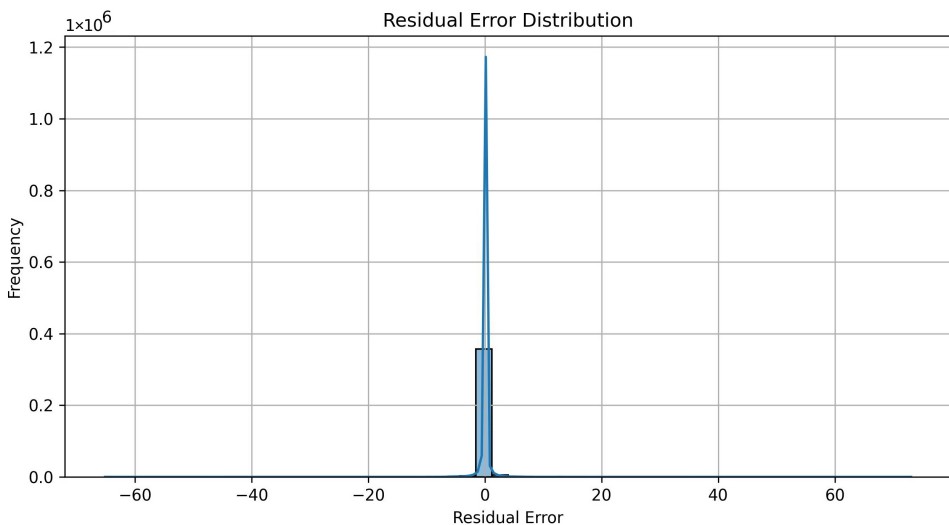

**Figure 7.** Residual error distribution.

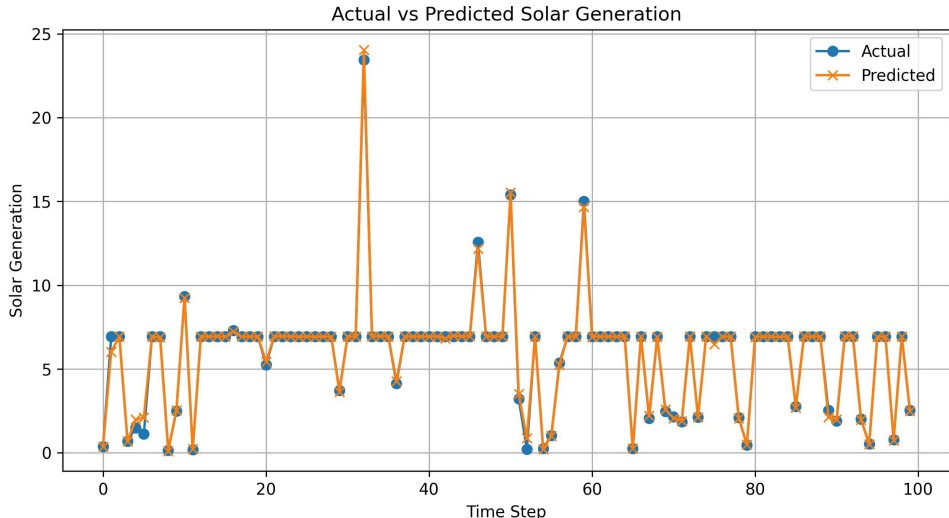

**Figure 8.** Solar energy generation vs. timestamp.

In the context of energy generation forecasting, the XGBoost Regressor optimized using Bayesian Optimization demonstrated exceptional predictive capability, as reflected in Table 4. With a low Mean Absolute Error (MAE) of 0.1854, Mean Squared Error (MSE) of 0.8999, and Root Mean Squared Error (RMSE) of 0.9486, the model consistently produced predictions that were closely aligned with actual generation values. Most notably, it achieved an $R^2$ score of 97.45%, indicating that the model effectively explains the vast majority of variance in the target data. This high $R^2$ value serves as a strong indicator of model accuracy, much like a high Area Under the Curve (AUC) in classification problems. Together, these metrics confirm that the XGBoost Regressor is a highly reliable and accurate tool for energy generation prediction.

**Table 4.** Performance metrics of XGBoost regressor with Bayesian optimization.

| Metric | Value |
|---|---|
| Mean Absolute Error (MAE) | 0.1854 |
| Mean Squared Error (MSE) | 0.8999 |
| Root Mean Squared Error (RMSE) | 0.9486 |
| $R^2$ Score | 97.45% |

*3.3. Real-Time API Integration for Energy and Weather Data*

The Flask API was developed to retrieve real-time weather conditions, process the ML-based energy predictions, and update the Ethereum smart contract.

A test request was made using curl to validate the API response:

The API request used for testing is shown in Listing 3. The API successfully processed the request and returned real-time weather and energy prediction data as shown in Figure 9.

**Listing 3.** Curl request to test the Flask API for real-time weather and ML-based energy predictions.

```
curl -X POST http://127.0.0.1:5000/predict \
-H "Content-Type: application/json" \
-d "{\"lat\":11.0168,\"lon\":76.9558}"
```

As illustrated in Figure 9, the latitude 11.0168° and longitude 76.9558° correspond to Coimbatore, India, a region with high solar energy potential. The response data reveal a Global Horizontal Irradiance (GHI) of 1039.23 W/m$^2$, indicating strong solar availability. The predicted energy generation of 1.068 kWh demonstrates the ML model's accuracy in forecasting production. Weather conditions are retrieved dynamically from the OpenWeather API, allowing for real-time pricing updates on the blockchain based on current environmental factors.

```
{
    "ghi": 1039.23,
    "predicted_energy": 1.068,
    "timestamp": "2025-03-16 10:50:32",
    "weather": {
        "AirTemperature": 30.78,
        "CloudOpacity": 0.4,
        "RelativeHumidity": 51,
        "WeatherCondition": 1,
        "WindSpeed": 2.57
    }
}
```

**Figure 9.** Weather conditions and energy prediction.

*3.4. Blockchain Contract Updates*

To verify that the contract is being updated correctly, an Alchemy Web3 API call was executed to fetch the most recent energy price stored on-chain.

The smart contract successfully returned the base price on the Ethereum network (see Listing 4).

**Listing 4.** Example output showing successful connection to the Ethereum network and the base price returned by the smart contract.

```
Successfully connected to the Ethereum
network.
Base Price: 217920817994 wei
```

This confirms that the contract is correctly linked to the Alchemy Web3 API, ensuring smooth interaction with Sepolia Testnet, and the latest price update is successfully stored on-chain, proving end-to-end integration from data ingestion to smart contract storage.

### 3.5. SQL Database Logging for Predictions and Weather Data

To track historical pricing trends, all ML predictions and weather updates are stored in an SQL database (`solar_predictions.db`). This setup enables persistent storage with timestamps, ensuring that each prediction is recorded for future analysis.

Table 5 illustrates the most recent entries logged into the SQL database, showcasing three records of ML-predicted solar energy output alongside corresponding cloud opacity values. Each entry is timestamped, ensuring precise tracking of prediction history over time. For instance, at 12:48:38 on 16 March 2025, the predicted energy output was 0.1241 kWh with a cloud opacity of 0.4, indicating moderate cloud cover. Comparatively, earlier entries show a slight decrease in cloud opacity (0.2) paired with fluctuating energy predictions, such as a notably low value of 0.01 kWh at 12:07:08. This suggests a possible non-linear relationship between cloud opacity and solar output, underlining the importance of continuous data logging for trend analysis and model refinement. The structure of the database not only ensures historical data preservation but also supports in-depth examination of environmental factors affecting solar energy generation.

**Table 5.** Recent database records for machine learning predictions.

| ID | Timestamp | Pred. Energy (kWh) | Cloud Opacity |
|:--:|:--|:--:|:--:|
| 3 | 16 March 2025 12:48:38 | 0.1241 | 0.4 |
| 2 | 16 March 2025 12:29:23 | 0.1588 | 0.2 |
| 1 | 16 March 2025 12:07:08 | 0.01 | 0.2 |

### 3.6. Blockchain Transaction Logs and Verification

Blockchain verification was performed by analyzing transaction logs to confirm that pricing updates were successfully recorded on-chain. The seller's transaction was successfully processed and validated on the Sepolia Testnet, ensuring that the contract address accurately reflected the updated energy price. Minimal gas fees were consumed, making the updates cost-efficient while maintaining transparency. The blockchain state changes confirm that the smart contract executed updates as expected, demonstrating the seamless integration of machine learning-based predictions, real-time weather data retrieval, and smart contract execution on Sepolia. This system provides a fully decentralized and AI-driven energy pricing mechanism, enabling real-time market adjustments based on solar energy availability and demand.

### 3.7. Seller's Address Transactions

The blockchain-based peer-to-peer (P2P) energy trading platform records every transaction on-chain to ensure transparency, traceability, and immutability. These transactions include interactions between sellers and the smart contract, such as energy pricing updates, listings, and sales. By leveraging blockchain, each transaction remains permanently verifiable, ensuring a fair trading ecosystem.

Figure 10 shows transaction details, including timestamps, block numbers, gas fees, and unique transaction hashes. These records help ensure fair energy trading and track seller behavior. Each transaction demonstrates how the seller engages with the smart contract, listing energy, adjusting prices, and completing transactions. Gas consumption and transaction fees reveal the computational cost of these operations. Maintaining an immutable, decentralized record facilitates energy trading and fosters participant trust.

### 3.8. Static Pricing Analysis

Static pricing blockchain transactions provide reliable standards for analyzing token values and market trends. By capturing unchangeable pricing data as a foundation for economic analysis in decentralized markets, this approach aids researchers in understanding price stability and trading patterns for energy tokens.

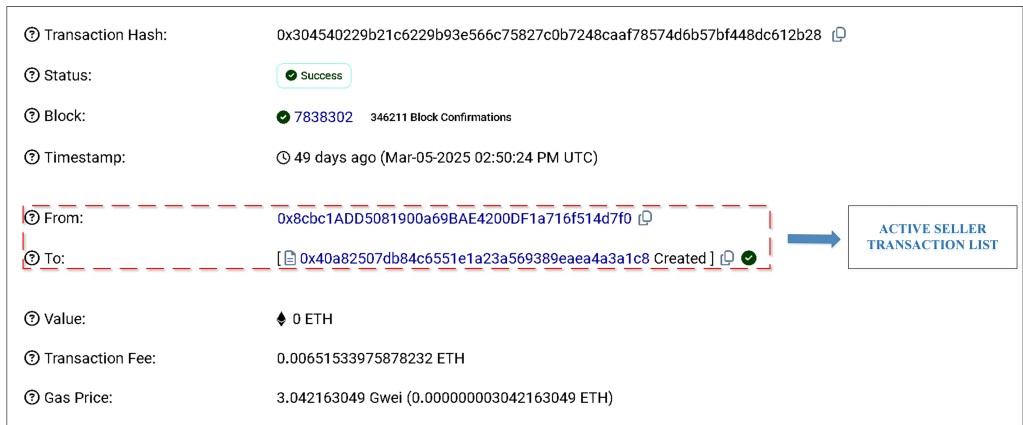

**Figure 10.** Seller's address transactions.

Figure 11 depicts an Ethereum transaction recording a transfer of five EnergyToken (ENGY) units on 4 April 2025, with 19 block confirmations. The transaction hash provides a unique identification, even though the timestamp (11:18:00 AM UTC) determines the execution time. This transaction implied creating a new contract address, therefore indicating potential expansion of new trading mechanisms within the energy token ecosystem. The static energy trading contract is the fundamental component of the peer-to-peer solar energy trading platform. By running on Ethereum Sepolia and being written in Solidity, it allows safe transactions at fixed prices between registered parties. The contract introduces EnergyToken (ENGY), which stands for energy units; the first supply is assigned to the administrator's wallet for ecosystem distribution.

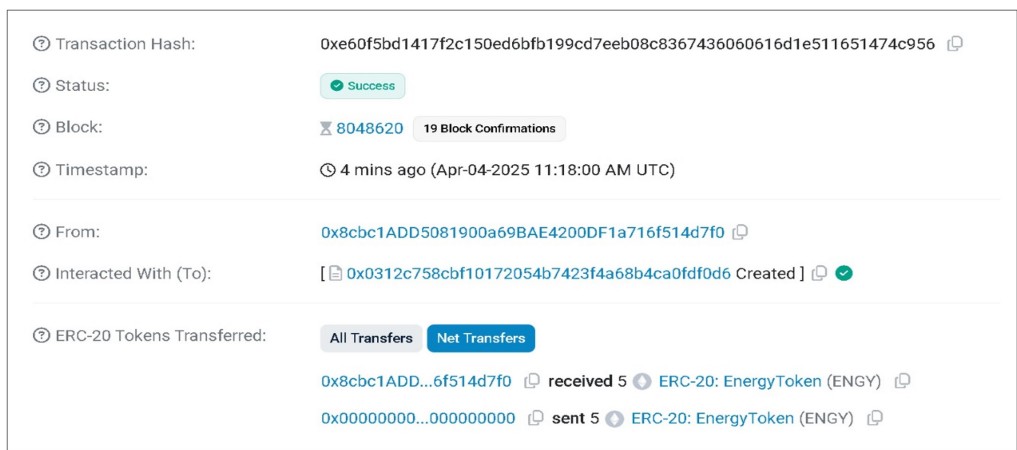

**Figure 11.** Ethereum transaction details showing EnergyToken (ENGY) transfer in static pricing.

Including OpenZeppelin libraries—ERC20, AccessControl, ReentrancyGuard—the contract strengthens security and role-based authorization. The ADMIN_ROLE manages price setting, token minting, and seller registration. Though the fixed price offers consistency, it overlooks solar irradiance affecting actual energy prices, weather changes, and real-time demand. The contract maintains a registry of active sellers under admin-registered addresses only eligible for token acquisition. The buyEnergy() function allows for safe

transactions and correspondingly transfers tokens by means of set pricing cost calculations. Openly, the contract produces events viewable via blockchain explorers: StaticPriceUpdated, SellerAdded, SellerRemoved, and Energy Purchased. Among the security elements are restricted minting powers and protection against readmission. Running on Sepolia, this is a lightweight version that uses very little gas and only requires 0.005–0.01 ETH. It makes no use of outside oracles or APIs. Unlike more complex dynamic solutions employing machine learning and real-time data, this simpler approach provides a clear starting point for tokenized energy transactions.

### 3.9. Dynamic Pricing-Based Energy Trading

The dynamic smart contract used in this work enables real-time energy pricing based on live data inputs. Operating on the Ethereum Sepolia testnet, this Solidity contract brings intelligence and flexibility to energy pricing using blockchain, Chainlink oracles, and ML projections. The pricing logic of the contract uses real-time grid prices, ML model forecasts (XGBoost with 97.45% $R^2$), current weather conditions, and time-of-day multipliers. The pricing engine calculates prices using Chainlink feed data and the ML model by adding weather and demand changes for three distinct time periods: Before Peak (09:00–18:00), Peak (18:00–22:00), and After Peak (22:00–09:00).

Figure 12 illustrates how the smart contract dynamically updates energy prices. The model integrates solar forecasts, weather conditions, and market demand to adjust rates efficiently, ensuring cost-effective trading while maintaining market stability. Security is maintained by role-based access control, which only lets the admin (DEFAULT_ADMIN_ROLE) see private functions. Rapidly turned away are those who try to enter uninvited for restricted activities. A system that guarantees every transaction gets its own unique nonce, thus preventing repeat attacks using nonces, helps to make the system more secure. Transactions trying to use the same nonce are rejected immediately. The contract lets the seller add or remove items and tracks them using indexed mapping. Among the events it sends on every change are PriceUpdated, MLPriceUpdated, WeatherConditionUpdated, and SellerAdded. It also provides a backup strategy should the Chainlink feed fail to run. These tools and tracking capabilities let frontends be open and observed. Using a minimally funded wallet of 0.6 ETH, the contract was tested on Sepolia, showing gas efficiency under real-time conditions, full functionality with live updates, and security against unauthorized tampering and widespread Ethereum attacks. This confirms the system's readiness for production deployment in energy trading markets requiring flexibility, security, and real-time adaptability.

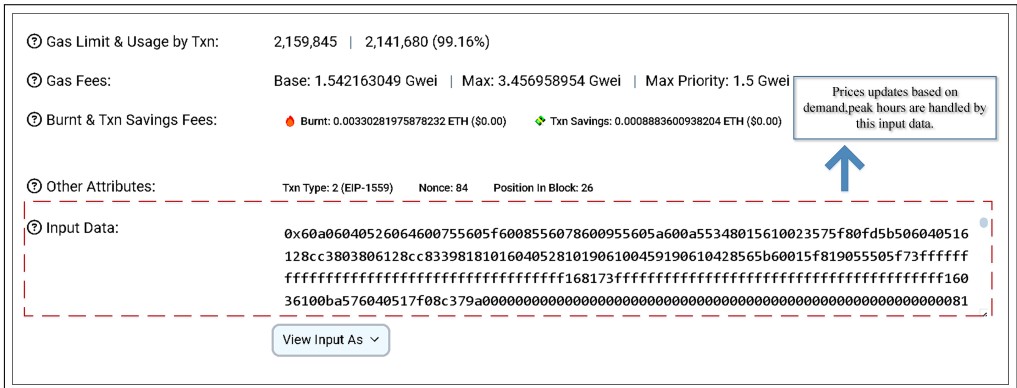

**Figure 12.** Dynamic pricing adjustments in the energy trading system.

### 3.10. Static Pricing vs. Dynamic Pricing

Static Pricing offers a set transaction fee, making it simple but inflexible during network congestion. On the other hand, dynamic pricing changes prices in real-time depending on demand, so improving efficiency and resource distribution.

Table 6 differentiates between two blockchain transactions under static and dynamic pricing models. Showing no gas metrics or dynamic features but rather token mining of three ENGY tokens, the static pricing transaction suggests a simple and pre-determined cost structure. The dynamic pricing transaction, on the other hand, runs a smart contract function call to record an active seller. This transaction emphasizes advanced features such as gas consumption monitoring, dynamic pricing set at 718 Gwei, and integration with an active seller tracking system.

**Table 6.** Comparison of static and dynamic pricing features.

| Feature | Static vs. Dynamic Value |
|---|---|
| Transaction Hash | 0x08c01296017417226 (Static) / 0x03d50258229c1f06 (Dynamic) |
| Status | Success / Success |
| Block Number | 9024004 / 9024008 |
| Timestamp | Apr 04, 2025, 11:18 AM UTC / Apr 05, 2025, 02:50 PM UTC |
| Sender Address | 0x91ce14d042081 / 0x80bd14d02801 |
| Receiver Contract | 0x011227438521 / 0x4ba52805984f |
| Transaction Type | Token Mining (Static) / Function Call (Dynamic) |
| Ether Transferred | 0 ETH / 0 ETH |
| Gas Used | Not Displayed / 410852 ETH |
| Gas Price | Not Displayed / 718 Gwei |
| Dynamic Pricing | Not Enabled / Enabled via `calculateDynamicPrice()` |
| Active Seller Tracking | Not Available / Via `getActiveSellers()` |
| ML-Predicted Price | Not Available / Via `mlPredictedPrice()` |
| Demand Multiplier | Not Included / Via `demandMultiplier()` |
| Weather Condition | Not Considered / Integrated via `updateWeatherAndDemand()` |
| Peak Hour Multipliers | Not Applicable / Implemented |
| External Oracle Usage | Not Supported / Chainlink Oracle for pricing and weather |
| Real-Time Data Handling | Static Only / ML and Oracle Based |
| Security Features | Basic Transfer / Role-based, reentrancy-safe |
| Analytical Value | Limited / Enables dynamic policymaking |

Because it employs several cutting-edge technologies that typical pricing systems lack, our proposed dynamic pricing approach much exceeds mere transaction processing. mlPredictedPrice() alters prices to fit to how a machine learning system projects the market will perform. Since the weather influences how much renewable energy is generated, the updateWeatherAndDemand() function facilitates environmental consideration.

The dynamic pricing model's main advantage is its improved resource allocation during periods of high demand. Demand multipliers and price variations during peak hours help to make these goals achievable. The system leverages Chainlink, a kind of outside oracle, to obtain constant grid pricing and meteorological data from dependable off-chain sources. Combining real-time external information in this manner allows for quicker and more accurate price choices. Dynamic pricing has significantly improved security and analysis. While gathering a great variety of data enables professionals to examine it so they may create plans and policies for reacting to demand, role-based access controls and reentrancy protection mechanisms keep deals safe. These qualities all enable us to be more explicit about the goals of our investigation.

Demand-responsive energy pricing allows the dynamic pricing system to find the best resource use in many demand situations. Improved peak-hour logic and real-time machine learning projections help to make this possible. Monitoring prices and changing them based on active sellers and dynamic demand multipliers suggests that network performance helps stabilize the grid. Clear choices for suppliers to register and track pricing will help users of the energy market to feel more secure and involved. Considering environmental factors and weather, price is a better predictor of the generation capacity of green energy sources whose output fluctuates with the circumstances. Thus, enhanced security, role-based access controls, and reentrancy prevention help keep the system stable and guarantee fair transactions. These characteristics offer a real-time trading system motivated by a combination of economic incentives, grid needs, and environmental objectives. This stops token abuse and energy waste.

Dynamic price systems complicated characteristics allow them quick and simple responses to network problems, market changes, and external factors. Dynamic pricing, smart contracts, transparency, and automation support scalable, flexible energy trading systems.

### 3.11. Transaction Event Logs

Transaction event logs ensure the accuracy and transparency of contract updates in our peer-to-peer energy trading platform. These logs record every interaction with the smart contract, including pricing updates, energy trading transactions, and modifications by active sellers. Figure 13 illustrates the documented event logs, serving as a permanent record of all executed functions.

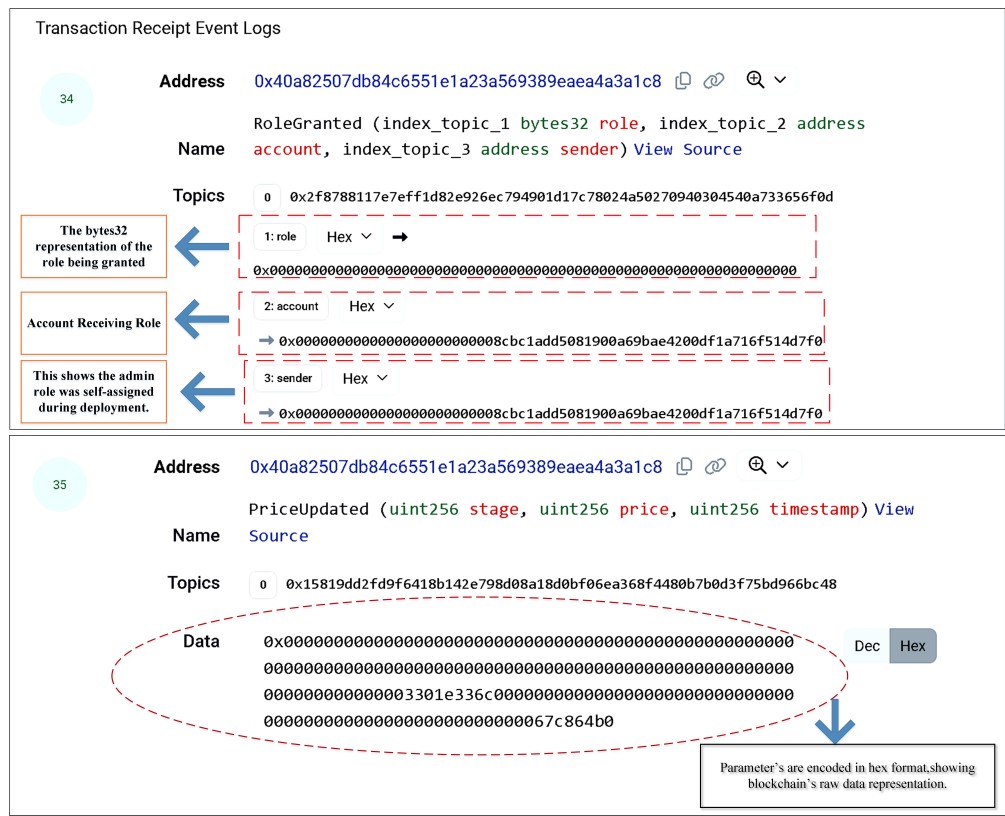

**Figure 13.** Transaction Event Logs in the smart contract.

The graphic presents two essential event logs from the smart contract. Event Log #34 documents a `RoleGranted` event that tracks permission allocations throughout the system. This log indicates the self-assignment of an administrator position during deployment, with three indexed topics: a bytes32 representation of the role, the account address obtaining

rights, and the sender address initiating the modification. This permission framework guarantees that only authorized individuals can alter the system.

Event Log #35 records a `PriceUpdated` event that monitors alterations in energy pricing data. This event comprises three parameters (`stage`, `price`, and `timestamp`), with all data encoded in a hexadecimal format, as shown in the raw data section. This thorough documentation of price modifications ensures pricing transparency for all network members.

These detailed event logs improve the system's openness, accountability, and dependability in several aspects. Initially, they furnish incontrovertible proof of all modifications to the contract's status. Secondly, they establish an audit trail for regulatory compliance by recording the individuals who implemented modifications and the corresponding timestamps. Third, they allow participants to independently check the authenticity of transactions and price adjustments. Ultimately, they function as definitive records for resolving disputes should conflicts occur among energy merchants. Our peer-to-peer energy trading platform fosters confidence among players and preserves the integrity of all energy transactions through a comprehensive logging mechanism.

### 3.12. Transaction State Verification

Verification of blockchain transaction states is essential for the secure and reliable execution of smart contract operations. This procedure is crucial for ensuring transparency and system integrity, since it verifies that all pricing updates and transactions are precisely documented on the blockchain.

As seen in Figure 14, the system authenticates and records modifications to smart contracts following each execution. This includes functions such as energy trading and dynamic pricing modifications. These alterations are documented on-chain, rendering them immutable. The solution guarantees accountability and adherence to the logic established in smart contracts by preserving a permanent and tamper-proof transaction history. This not only bolsters confidence among participants but also safeguards against unauthorized modifications and fraudulent actions.

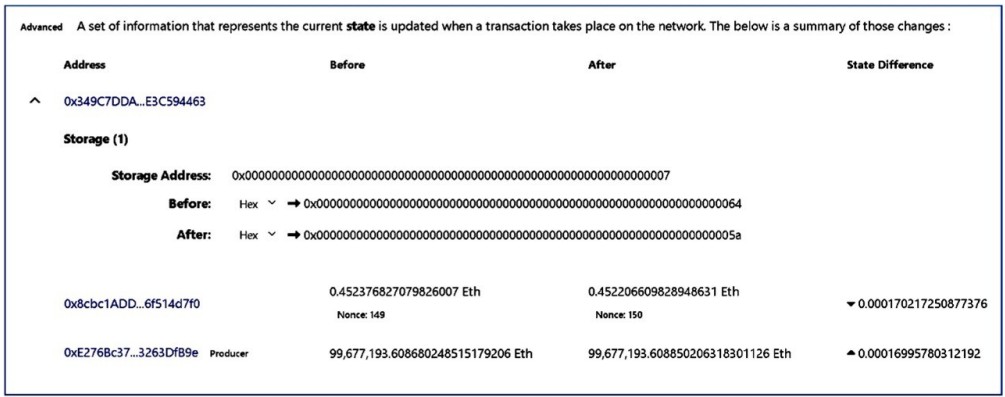

**Figure 14.** Transaction state verification in the Smart Contract.

### 3.13. Security Mechanisms and Unauthorized Access Handling

To ensure integrity and protection against unauthorized manipulation, only the admin (DEFAULT_ADMIN_ROLE) is permitted to invoke sensitive functions such as updateML-Price, setWeatherCondition, or add/remove sellers. The contract integrates role-based access control via OpenZeppelin's AccessControl. When an unauthorized account attempts to perform restricted actions (e.g., updating ML price), the system immediately rejects the request, preventing any gas wastage or state modification. This is evident from the backend logs shown in Figure 15.

The contract canceled the transaction when a regular wallet tried to change the ML price without enough money or permission, showing that prices were protected and access was controlled. The contract additionally establishes a mechanism to prevent replay attacks utilizing nonces. Each significant transaction possesses a unique nonce; any nonces that are utilized multiple times are promptly disregarded. Figure 15 illustrates the requisite nonce for executing and transmitting a legitimate transaction from the administrator. The contract rejected the replayed transaction data due to error code -32603, which involved the same nonce. This case demonstrates that the contract ensures each transaction is distinct and the timestamp is accurate, hence enhancing its resistance against spoofing and replay attacks.

```
☐ Connected to Sepolia
☐ Admin: 0x8cbc1ADD5081900a69BAE4200DF1a716f514d7f0
☐ Unauthorized: 0x111fA8C4A7b0480F1ad89815a592EdAf7bc78940
☐ Contract nonce: 3

☐ Attempting unauthorized ML price update...
☐ Unauthorized Access Blocked: {'code': -32003, 'message': 'insufficient funds for gas * price + value: have 0 want 1000000000000000'}

☐☐ Attempting replay attack...
☐ Valid admin tx sent: a29089fa41fb7e92f270c1a56809080554a0a05370a2c603afd60bb3cedf372e
☐ Replay Attack Blocked: {'code': -32603, 'message': 'replacement transaction underpriced'}
```

**Figure 15.** Security monitoring logs showing blocked unauthorized access attempts and replay attack prevention.

## 4. Conclusions

This paper demonstrates the development of a peer-to-peer blockchain solar energy trading system utilizing Ethereum's Sepolia testnet, integrating real-time meteorological data, machine learning forecasts, and dynamic pricing mechanisms. The suggested XG-Boost model attained an accuracy of 97.45% in predicting solar output, enabling precise estimation of energy generation depending on climatic conditions. The implemented Solidity smart contract facilitates energy trading via dynamic price modifications influenced by supply and demand, peak times, and projected output. Experimental results confirm system effectiveness through efficient contract execution, transaction verification, and automatic pricing modifications. The use of the Wei denomination for pricing enables precise microtransactions and minimizes gas costs. Database integration ensures comprehensive recording of all predictions and transactions, hence guaranteeing total system transparency.

Despite the promising results, the system faces several limitations that constrain its immediate large-scale deployment. The current implementation operates on the Ethereum Sepolia testnet with inherent scalability constraints of approximately 15 transactions per second, which may pose challenges for high-volume energy markets. The reliance on external weather APIs introduces potential points of failure during network outages or service disruptions. The machine learning model requires continuous retraining to maintain predictive performance as weather patterns evolve, and the current focus on solar energy trading may not fully address multi-source renewable energy portfolios. Future work should contemplate the integration of Layer-2 scaling solutions such as Polygon or Arbitrum to improve transaction throughput, implementation of cross-chain interoperability protocols to expand market liquidity, development of multi-energy source prediction models incorporating wind and storage systems, integration of IoT sensors for direct energy meter readings, carbon credit trading mechanisms, and regulatory compliance frameworks for legal deployment across different jurisdictions.

This research illustrates that the amalgamation of machine learning with blockchain technology facilitates an open, secure, and efficient autonomous energy market. The findings may accelerate the shift towards more ecologically sustainable energy solutions.

**Author Contributions:** Conceptualization, J.N.B., C.N. and C.R.; methodology, J.N.B., A.P. and C.N.; software, D.P., R.J.B. and A.P.; formal analysis, A.P. and R.M.R.Y.; investigation, J.N.B., R.D.A.R. and C.R.; resources, R.D.A.R. and R.M.R.Y.; data curation, D.P., R.J.B., A.P. and R.D.A.R.; supervision, C.N. and C.R.; project administration, C.R. All authors have read and agreed to the published version of the manuscript.

**Funding:** This research received no external funding.

**Institutional Review Board Statement:** Not applicable.

**Informed Consent Statement:** Not applicable.

**Data Availability Statement:** The datasets used in this study are publicly available and can be accessed through the original source cited in the manuscript. No new data were generated in this study.

**Acknowledgments:** The authors thank their respective institutions for providing computational resources and infrastructure. No external administrative or technical support was involved.

**Conflicts of Interest:** The authors declare no conflicts of interest.

# Nomenclature

| Symbol | Description |
| --- | --- |
| $X$ | Feature matrix for ML prediction ($\mathbb{R}^{n \times d}$) |
| $T_t$ | Air temperature at time $t$ (°C) |
| $I_t$ | Solar irradiance at time $t$ (W/m$^2$) |
| $W_t$ | Wind speed at time $t$ (m/s) |
| $H_t$ | Humidity at time $t$ (%) |
| $C_t$ | Cloud cover percentage at time $t$ (%) |
| $P_{t-k:t-1}$ | Historical energy generation from $t - k$ to $t - 1$ (kWh) |
| $\Delta T_t$ | Temperature gradient at time $t$ |
| $h$ | Hour of day (0–23) |
| $\sin(2\pi h/24)$, $\cos(2\pi h/24)$ | Trigonometric variables for time-of-day patterns |
| $n$ | Number of samples |
| $Y_i$ | True value for data point $i$ |
| $\hat{Y}_i$ | Predicted value for data point $i$ |
| $\overline{Y}$ | Mean of $Y_i$ ($\overline{Y} = \frac{1}{n} \sum_{i=1}^{n} Y_i$) |
| $\theta$ | ML model parameter/hyperparameter vector |
| $\Theta$ | Parameter search space |
| $\theta^*$ | Optimal parameter vector |
| $E[L(\theta)]$ | Expected loss function |
| $L(\theta)$ | Loss for parameters $\theta$ |
| $K$ | Number of folds in cross-validation |
| $N_k$ | Samples in fold $k$ |
| $D_k$ | Validation dataset in fold $k$ |
| $MAE$ | Mean Absolute Error |
| $MSE$ | Mean Squared Error |
| $RMSE$ | Root Mean Squared Error |
| $MAPE$ | Mean Absolute Percentage Error |
| $R^2$ | Coefficient of determination |
| $\eta$ | Learning rate (XGBoost) |
| $d$ | Maximum tree depth (XGBoost) |
| $\lambda, \alpha$ | L2, L1 regularization coefficients (XGBoost) |
| $P_t^{base}, P_t^{final}$ | Base price and final dynamic price at time $t$ |
| $S_t, D_t$ | Total supply and demand at time $t$ (kWh) |
| $ENGY$ | EnergyToken (ERC-20 token) |

| *RBAC* | Role-Based Access Control |
| Chainlink Oracle | Oracle for real-time blockchain data |
| Sepolia Testnet | Ethereum test network |

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
