# Peer review of "Blockchain-Enabled Secure Energy Transactions for Scalable and Decentralized Peer-to-Peer Solar Energy Trading with Dynamic Pricing"

_technologies, doi:10.3390/technologies13100459_

Round 1
Reviewer 1 Report
Comments and Suggestions for Authors
This paper presents a machine learning–driven, Ethereum-based decentralized solar energy trading platform that integrates accurate XGBoost forecasting, dynamic pricing via smart contracts, and secure oracle-based data feeds to enable real-time, adaptive, and trustless energy markets. Overall, the comparisons are sufficient and interesting. However, there are several points where clarifications, justifications, and expansions would improve the manuscript.
- Renewable energy generation prediction is a central focus of this study. However, the introduction does not adequately discuss renewable energy generation. For instance, abnormal data may affect prediction accuracy and should be identified and removed from the dataset. In this context, the study “managing massive res integration in hybrid microgrids: a data-driven quad-level approach with adjustable conservativeness” should be cited and compared in Table 1 to strengthen the introduction.
- In addition, many advanced prediction techniques, such as Transformer models, have been applied to renewable energy forecasting. Why was the XGBoost model chosen instead?
- It is recommended to provide a detailed description of the XGBoost regressor model used for renewable energy generation prediction.
- In the case study results, it is suggested to include a quantitative comparison and discussion with existing methods.
- Some figures, such as Figures 3 and 9, are unclear. It is recommended to replace them with higher-resolution versions.
- The conclusion should also address the limitations of the proposed method and outline potential directions for future research.
Reviewer 2 Report
Comments and Suggestions for Authors
The article "Blockchain-Enabled Secure Energy Transactions for Scalable and Decentralized Peer-to-Peer Solar Energy Trading with Dynamic Pricing" was reviewed. I have the following observations:
- Include the novelty of this study in the abstract.
- Before the introduction, it is important to include at least one paragraph that allows the reader to contextualize the topic of interest. Additionally, before presenting Table 1, begin by explaining and addressing the details of the content. In the same introductory section, include the research objective, questions, and key contributions.
- Equations 1-15 are clearly numbered and referenced in each preceding paragraph. However, it is important to indicate the parameters involved in each one and their technical details.
- It is important to create a section reviewing previous work. There are no references that contextualize the topic; please address this point. It is recognized that Table 1 was created in the introduction and that some references exist, but it is insufficient.
-The conclusions could be much more compelling based on the research conducted; it is appropriate to provide additional conclusions and clarify some limitations identified in this study.
Please address these observations for further review and decision-making.
Reviewer 3 Report
Comments and Suggestions for Authors
This paper presents a machine learning–based solar energy trading platform on the Ethereum blockchain. The topic is interesting, however, the reviewer has the following comments to improve the quality of manuscript.
1. The introduction section should clearly highlight the research gap and key contributions.
2. Nomenclature section should be added in the manuscript.
3. How does the presented framework ensure the socio-economic welfare of market participants.
4. The test system for validating the proposed model is not clear.
5. Authors should include the future work for the manuscript.
Round 2
Reviewer 1 Report
Comments and Suggestions for Authors
Thanks for the careful revisions. The authors have fully addressed all my concerns.
Reviewer 2 Report
Comments and Suggestions for Authors
A new review of the article "Blockchain-Enabled Secure Energy Transactions for Scalable and Decentralized Peer-to-Peer Solar Energy Trading with Dynamic Pricing" was conducted. I believe that most of the requested changes have been fully addressed and meet quality standards. Therefore, I suggest its publication.